# Revolutionizing Eye Care: Exploring the Potential of Microneedle Drug Delivery

**DOI:** 10.3390/pharmaceutics16111398

**Published:** 2024-10-30

**Authors:** Satish Rojekar, Swapnali Parit, Amol D. Gholap, Ajit Manchare, Sopan N. Nangare, Navnath Hatvate, Vrashabh V. Sugandhi, Keshav Raj Paudel, Rahul G. Ingle

**Affiliations:** 1Department of Pharmacological Sciences, Icahn School of Medicine at Mount Sinai, New York, NY 10029, USA; 2Institute of Chemical Technology, Marathwada Campus, Jalna 431203, India; swapnaparit2012@gmail.com (S.P.); j22pht.ab_manchare@stumarj.ictmumbai.edu.in (A.M.); navnath711@gmail.com (N.H.); 3Department of Pharmaceutics, St. John Institute of Pharmacy and Research, Palghar 401404, India; amolgholap16@gmail.com; 4Department of Pharmaceutics, H. R. Patel Institute of Pharmaceutical Education and Research, Shirpur 425405, India; snangareopan@gmail.com; 5College of Pharmacy & Health Sciences, St. John’s University, 8000 Utopia Parkway, Queens, NY 11439, USA; rushabhsugandhi@gmail.com; 6Centre for Inflammation, School of Life Sciences, Faculty of Science, Centenary Institute and University of Technology Sydney, Sydney, NSW 2007, Australia; keshavrajpaudel@gmail.com; 7Datta Meghe College of Pharmacy, Datta Meghe Institute of Higher Education and Research (Deemed to Be University)—DMIHER, Wardha 442107, India

**Keywords:** eye care, glaucoma, biocompatibility, microneedles, ocular drug delivery

## Abstract

Microneedle technology revolutionizes ocular drug delivery by addressing challenges in treating ocular diseases. This review explores its potential impact, recent advancements, and clinical uses. This minimally invasive technique offers precise control of drug delivery to the eye, with various microneedle types showing the potential to penetrate barriers in the cornea and sclera, ensuring effective drug delivery. Recent advancements have improved safety and efficacy, offering sustained and controlled drug delivery for conditions like age-related macular degeneration and glaucoma. While promising, challenges such as regulatory barriers and long-term biocompatibility persist. Overcoming these through interdisciplinary research is crucial. Ultimately, microneedle drug delivery presents a revolutionary method with the potential to significantly enhance ocular disease treatment, marking a new era in eye care.

## 1. Introduction

### 1.1. Background

Ophthalmic conditions are classified into two distinct segments: segments of barriers of the eye are classified as the anterior segment (encompassing the cornea, conjunctiva, lens, and ciliary body) and the posterior segment (comprising the sclera, vitreous humor, retina, choroid, and optic disc) [1]. Eye diseases and injuries cause severe visual impairment or blindness globally [2]. Addressing conditions impacting the rear portion of the eye, like diabetic retinopathy, retinitis pigmentosa, retinoblastoma, and choroidal neovascularization (CNV), poses significant hurdles due to ocular barriers’ complex structure and function [3]. A wide range of conditions can cause severe vision impairment, including uncorrected refractive errors, retinopathy, allergies, conjunctivitis, dry eye syndrome, scleral and iris disorders, cataracts, glaucoma, central retinal vein occlusion, and diabetic macular edema [4,5]. Cataracts are the primary cause of global blindness, constituting approximately 51% of cases. Although predominantly affecting older adults, they can develop in younger individuals due to genetic predisposition, trauma, or medication use, as presented in Figure 1. The prevalence of cataracts escalates with age, with more than half of Americans aged 80 or older having either undergone cataract surgery or experiencing cataract formation [6]. Glaucoma encompasses a collection of eye diseases marked by optic nerve damage, frequently linked to elevated intraocular pressure (IOP), and ranks as the world’s second leading cause of blindness. Its occurrence rises with age, affecting around 3.54% of individuals aged 40–80 globally, with higher occurrences among older age brackets [7]. Age-related macular degeneration (AMD) is a prominent reason for permanent vision impairment amongst older adults, impacting the central portion of the retina and the macula and resulting in blurred or distorted central vision. Its prevalence escalates with age, particularly affecting individuals over 50 years old, with more than 196 million people worldwide impacted by the condition [8]. Diabetic retinopathy emerges as a complication of diabetes, impacting the retina’s blood vessels and ranking among the primary reasons for blindness in working-age adults. Its prevalence aligns with the duration of diabetes, affecting about one-third of individuals with the condition, with the risk amplifying over time [9].

Delivering drugs to the eye poses significant challenges because of ocular tissues’ highly fragile, relatively inaccessible, and barrier-rich properties. Ophthalmic diseases and disorders encompass various eye and vision conditions [10]. Traditional therapies for these conditions commonly encounter numerous obstacles. Eye drops present several drawbacks, such as limited bioavailability (<5%), the drainage through the nasolacrimal system, systemic absorption, and lymphatic drainage, making it challenging to attain and withstand therapeutic concentrations at the retina through this route [11]. Ophthalmic preparations, such as gels, ointments, or eye drops, are commonly used for eye conditions. However, they require regular dosing, which can lead to treatment plan non-adherence, reducing therapy effectiveness. They are applied to the conjunctival sac, the eye’s surface, or the eyelid by medical professionals or the patient. It can lead to non-adherence to treatment plans, reducing therapy effectiveness. As a result, in addition to creating the vehicle/base composition, medications are often integrated into suitable carriers or systems designed to supply the required concentration in the treated tissue for the intended duration [12]. 

They are traditionally applied to the cornea, sclera, or suprachoroidal space (SCS), allowing for drugs to overcome barriers. Ocular drug delivery has three conventional approaches, topical application, intraocular injection, and systemic administration, with certain drawbacks when effectively delivering medication to the posterior segment of the eye [13]. Other avenues for administering drugs to the eye involve surgically implanting drug carriers to enable prolonged drug release into ocular or periocular tissues and precise topical administration through injections and conventional topical applications [14,15]. Nanotechnology could improve ocular therapy by addressing issues like poor intraocular penetration and rapid ocular elimination in traditional drug delivery routes [10]. The microneedle technique has been investigated as a promising method to enhance eye treatment, especially with coated, dissolving, and hollow types, proving particularly effective in drug delivery [16]. Conditions like myopia and presbyopia can often be corrected with glasses or lenses. In contrast, more severe conditions like age-related macular degeneration (AMD) and diabetic retinopathy require extensive treatment, posing a burden on healthcare systems.

Chronic ocular diseases necessitate long-term treatment, with anterior segment diseases typically managed through topical drug delivery and posterior segment diseases through intraocular injections [11].

The literature indicates that topical drug administration for ocular conditions typically results in only 5–10% of the total dose reaching the target tissues due to non-productive absorption by the conjunctiva or systemic drainage. This necessitates higher doses, leading to potential ocular toxicity. Oral drugs face solubility and permeability issues, often failing to reach therapeutic levels at the target site [17]. Microneedles offer a minimally invasive approach, penetrating only a few hundred microns into the sclera to avoid damage to deeper ocular tissues. They enable the deposition of drugs or drug carriers into the sclera or the suprachoroidal space, facilitating drug diffusion into deeper ocular tissues like the choroid, retina, and vitreous humor. Researchers are exploring the potential of microneedles for drug delivery across various routes, including the eye, to enhance therapeutic outcomes while minimizing invasiveness. Ongoing developments in microneedle technology are being investigated for their applicability and effectiveness in ocular drug delivery [18].

The ocular drug delivery system is suffering from challenges owing to physiological processes like blinking and nasolacrimal drainage, efflux pump, anatomical barriers, and metabolism into the ocular tissues, favoring drug elimination. The mucin layer on the eye prevents exogenous substances from permeating the deeper tissues. A longer duration of pharmaceutical therapy is usually required to manage inflammation and proliferative ocular diseases effectively. Overcoming the ocular barriers is a major issue for traditional treatment tools like eye ointments and drops, restricted bioavailability, and the need for frequent administration. The ocular barrier can be countered by direct injection into the ocular tissue. The injection poses a risk of tissue injury and consists of unfavorable side effects resulting in poor patient compliance [11,19,20]. 

Zero-order kinetics is involved in the diffusion of the drug substance through a constant controlled rate for ophthalmic therapeutic systems like non-biodegradable inserts. Another non-invasive method for ophthalmic administration of drugs delivers the drug mostly by injection to the vitreous or sub-surface parts of the eye. Retrobulbar injections and subconjunctival injections can provide immediate or prolonged drug release based on the composition of the formulation. Some of the leading disadvantages include local toxicity, tissue damage, optic nerve injury, eyeball perforation, occlusion of the central retinal artery or vein, and direct retinal toxicity during the accidental puncture of muscles. On the other hand, intravitreal injection is one of the alternatives involved in painless procedures but suffers from the limitation of the recall of drug action in case of side effects or toxic effects, including retinal inflammation. The minimization of tissue damage, reduction in membrane continuity disruption, and elimination of the risk of pathogen infections with overall safety requires controlled drug release with minimization into the same needle size. 

Microneedles are emerging as the promising delivery technology for the administration of medication to eye conditions to provide accurate, less invasive, and localized medication administration for ocular diseases [12,21]. 

### 1.2. Microneedle Technology

Microtechnology is quickly making its way into the field of pharmaceutical sciences, particularly pharmaceutical technology, after initially emerging in biomedicine. The remarkable advancement in new manufacturing techniques presents prospects for developing exact and complex drug delivery instruments [12,22]. The microneedle platform features an innovative drug delivery system comprising miniature-sized needles [23,24]. Sustained ocular drug delivery has garnered significant attention in recent times to supplant the need for frequent intravitreal injections. Treatments for eye diseases, including eye drops or ointments, are frequently intrusive. Overcoming obstacles, microneedles are a revolutionary delivery technology that provides localized, efficient, and less intrusive drug administration to the eye, offering promising health effects [25,26]. Microneedles are minimally invasive tools designed for targeted and extended drug delivery to address chronic ailments. The Food and Drug Administration (FDA) states that microneedling devices produce numerous small puncture holes in the skin. Given the delicate nature of the eye, microneedle administration poses challenges and complexities for drug delivery at this site [11,27]. The transdermal microneedle structure and the “Poke and Patch” mechanism of solid microneedle are presented in Figure 2.

Recent research has confirmed the potential benefits of microneedles in facilitating drug delivery systems (DDSs) located inside targeted ocular tissues. The delivery of formulations to the eye can be altered using microneedles [28]. In 1905, Dr. Ernst Kromayer documented the initial use of microneedles, proposing using motorized dental burs to treat scarring and hyperpigmentation. However, it was in the 1960s that the concept of drug delivery via microneedle platforms garnered significant attention [11]. Silicon is the first material used to produce microneedle arrays due to its versatile properties and capability to form various microneedle geometries. The most often utilized materials in the manufacturing of microneedles contain metals like stainless steel and ceramics, titanium, and silicon. Furthermore, non-biodegradable polymers like photolithographic epoxy resins are employed, as well as biodegradable polymers like poly(lactic-co-glycolic acid (PLGA), polyglycolic acid (PGA), and polylactic acid (PLA). Different shapes and sizes for various applications are available [29]. Studies highlight hydrophilic matrices made from polyvinylpyrrolidone (PVP) and hyaluronic acid for microneedles in ocular drug delivery, demonstrating their ability to produce mechanical characteristics and rapid drug release for therapeutic effectiveness and patient comfort [12]. Experimental research has been directed at the usage of microneedles in the context of targeted and localized drug administration [30]. As mentioned in this review, hollow, solid, and dissolving microneedles are amongst the microneedle varieties used for ocular applications. In essence, microneedles hold the potential to revolutionize drug delivery by facilitating improved targeting and localization, especially for medications that pose challenges when administered through conventional methods [11].

While it concerns eye-related operations, using microneedles, which are shorter than 1 mm, presents a less intrusive option than using conventional hypodermic needles, which have more than 10 mm lengths for intraocular injections. This lessens tissue damage and permits more targeted, tissue-specific medication delivery. The unique characteristics of DDSs based on microneedles confer several benefits when compared to alternative techniques utilized for the administration of drugs into the eyes. Their special and beneficial characteristics are further enhanced by the various routes that can deliver microneedles to the eye [31]. Microneedles are designed to reduce discomfort and potential adverse effects, including irritation, infection, tissue damage, and inflammation. These formulations are designed for brief stays on the eye’s surface [12]. Injectable formulations can precisely transport the necessary amount of drug to the targeted eye area; they are the most significant novel drug delivery technologies. As advancements in single-microneedle technologies continue, simultaneous research aims to develop microneedle systems and patches for administering ocular medication [32]. Another important advantage of microneedle-based devices over intravitreal injections for treating posterior segment illnesses is localized medication delivery. Medication micro-depots that dissolve microneedles in the ocular layers enable continuous medication release. The polymeric elements of dissolving microneedles serve as substrates for sustained drug release, thereby promoting drug dissolution and continuous release from these reservoirs over an extended period [11]. This enhances patient acceptance and reduces the clinical burden associated with disease treatment. It has advantages over traditional formulation tactics, such as topical eye drops. Microneedles provide therapeutic benefits in addition to clinical ones. Their reduced size compared to hypodermic needles may help prevent needle-phobia problems related to intravitreal injections and other operations [33]. Additionally, the excellent resolution of 3D printing makes it possible to employ microneedles in applications requiring dimensional accuracy and having a low tolerance for dimensional mistakes, like vascular tissues or the eyes [34].

### 1.3. Purpose of the Review

This review discusses the challenges and lessons learned from microneedle research for ocular applications, focusing on their background, benefits, and the current state of study. This review aims to draw attention to these difficulties and provide insight into the lessons that may be applied from current microneedle research to facilitate the clinical transformation of these platforms for ocular applications. Consequently, there remains a lot of potential for improvement in utilizing finite element simulation in developing microneedles, especially for ocular applications. The use of microneedles in eye therapy has become more prevalent. The most suitable treatment depends on the ocular disease’s location and underlying cause. Microneedles are used as a novel delivery system for ocular pharmacological agents, including anti-vascular endothelial growth factor (VEGF), anti-inflammatory, and antiglaucoma agents, to treat ocular diseases like neovascularization and inflammation. The type of microneedle used for treatment depends on the intended application. Dissolving microneedles are suitable for eye diseases, as they can be applied like contact lenses, improving patient acceptability. Hollow and solid microneedles are suitable for posterior segment diseases, requiring precise administration procedures [11]. Microneedles offer precise and consistent outcomes with minimal inter-subject variability in bioavailability. Despite their numerous advantages, they do present some constraints: Potential skin irritation or allergic reactions may occur, especially in sensitive skin. Due to their tiny and thin size associated with hair thickness, microneedle tip breakage is possible, which could lead to complications if left in the skin. However, these constraints are infrequent and can be mitigated by employing advanced materials to select microneedles. Microneedles provide precise outcomes, enhanced therapeutic benefits, and low variability in bioavailability. Nevertheless, they have drawbacks, such as the potential for allergies and skin irritation. Sophisticated material selection can overcome these restrictions. This study explores the benefits and disadvantages of various microneedle types for ocular drug delivery applications. Microneedles offer a slightly invasive, tissue-specific drug delivery method, offering advantages over conventional hypodermic needles due to their unique features. Microneedles provide a significant advantage by overcoming physiological barriers. Microneedle-based devices provide a considerable advantage over intravitreal injections in localized drug delivery for treating posterior segment diseases [31]. Dissolving microneedles produce drug micro-depots in the targeted ocular tissue’s eye layers, allowing for sustained drug release by polymeric components, offering advantages over conventional formulation strategies. Dissolving microneedles have proven effective in showcasing the feasibility of prolonged large-molecule release, such as biologics, within the sclera. Additionally, microneedles offer therapeutic benefits beyond their clinical implications. Their reduced size compared to hypodermic needles may help prevent needle-phobia problems related to intravitreal injections and other procedures [35]. There are still several restrictions on dissolving microneedles. Due to the small dimensions of the needles (about less than 1 mm in height) and the therapy only being located in the needles, there is restricted loading capacity. Ocular microneedle research focuses on administering potent drugs to achieve therapeutic levels, while patch size is limited by eye curvature, affecting needle insertion effectiveness. It is challenging to recreate physiological conditions [36]. Coated microneedles in ocular applications have limitations, such as limited loading capacity, frequent administration, variable drug release rates, and poor repeatability due to coating process reductions in needle sharpness, resulting in suboptimal treatment of chronic ocular diseases and suboptimal insertion and delivery efficiency [37]. Securing the microneedle patch in place presents a notable challenge in ocular drug delivery. Even a well-designed microneedle patch intended for sustained drug delivery would lose its effectiveness if it were to detach from the application site. To use microneedles for continuous ocular drug administration, it is crucial to consider the importance of securing the microneedle patch in place [38].

## 2. Microneedle Design and Fabrication

### 2.1. Types of Ophthalmic Microneedles

Different types of ophthalmic microneedles, the materials used, and fabrication methods are listed in Table 1. The microneedle material selection and its characteristics are also listed in Table 2.

#### 2.1.1. Solid Microneedles

Due to their simplicity and widespread use, solid microneedles have been the primary choice for early research on microneedle drug or vaccine delivery. They are employed as a skin pretreatment, producing temporary micron-sized channels through the stratum corneum and mechanically distorting the epidermis before drug administration. However, solid microneedles alone cannot distribute or facilitate the passage of drugs [39,40]. Then, medications were injected or applied directly to the skin region punctured by microneedles in square patches. Scientists described a method involving the construction of a device utilizing a nanoscale zinc oxide pyramidal rod array. The drug release mechanisms through different types of microneedle patches are presented in Figure 3a, and the mechanism of the different kinds of microneedle drug release is presented in Figure 3b.

An array a previous study consisted of rods measuring 50 µm in length, with tip and base diameters of 60 nm and 150 nm, respectively [41]. These microscopic needles have sturdy, pointed structures that pierce the skin and leave behind tiny channels. Metals, polymers, or silicon are just a few materials that can create solid microneedles. They are frequently applied to extract interstitial fluid for analysis and drug delivery [42]. Solid microneedles composed of biodegradable polymers demonstrate ample mechanical robustness to breach the stratum corneum, thereby augmenting the effectiveness of PLA microneedles and enhancing drug delivery efficiency. Microneedles with an 800 µm depth and 256 microneedles per cm density were discovered to improve drug permeation. Researchers from various fields have also studied stainless steel microneedles. After employing stainless steel microneedle arrays, researchers investigated the improved delivery of captopril and metoprolol tartrate [11,12,43].

#### 2.1.2. Hollow Microneedles

Hollow microneedles, usually shorter (typically less than 1000 µm), have a similar basic structure to conventional hypodermic needles, featuring a hollow core through which the drug solution is delivered. Following hollow microneedles, the drug solution can be administered in either direction [44]. The drug solution can be administered actively, using a pressure-driven flow over the needle lumen, or passively, through diffusion. Occasionally, drug solutions may be injected into the viable epidermis, situated 60 to 130 µm below the stratum corneum. Hollow microneedles can be employed to create pores in the skin for constant delivery from a drug formulation deposited in a reservoir or for sampling body fluids. Hollow microneedles allow for deliberate regulation of drug flow rate using traditional flow-control tools such as syringes and micropumps [45]. Characterized by high molecular weights, proteins, antigens, and oligonucleotides are commonly utilized, and microneedles can effectively administer substantial doses of these drugs at a consistent flow rate. These tiny needles can be used to deliver drugs directly to their intended recipients while minimizing drug waste. In addition to their applications in signal monitoring and blood and tissue sampling, they have certain drawbacks, such as the needle becoming blocked after insertion into the skin. Materials such as silicon, metal, glass, polymers, and ceramic can create hollow microneedles [46]. Furthermore, force drugs can be delivered via pressure-driven force by integrating a microneedle injection applicator with a syringe pump or electromagnetic applicators. Patients’ preferences for better dosage control can be met. Moreover, hollow microneedles can include a micropump, microfluidic chip, or heater to deliver medications to the skin in a controlled manner [47]. Norman et al. examined the precision and reliability of standard hypodermic syringes (employing the Mantoux technique) [48], hypodermic needle adapters, and hollow microneedles for intradermal injection into pig skin. The percentage of drugs administered using each method showed similar levels of reliability (95.4 ± 4.9%, 97.6 ± 1.5%, and 94.9 ± 0.3%, respectively). Additionally, accuracy, measured as the proportion of the dose concentrated in the dermis, was comparable, at 97 ± 16%, 92 ± 21%, and 99 ± 12%, respectively [44,48].

#### 2.1.3. Dissolving Microneedles

Maltose is a structural framework for fabricating dissolving microneedles due to its ability to transition among three states—liquid, glassy, and solid—through precise temperature control during manufacturing. Maltose transitions into a liquid state above its melting point (Tm), and upon cooling below the glass transition temperature (Tg), the viscosity of the liquid maltose gradually rises, resulting in the formation of a solid state [49]. The triple state of maltose created the perfect conditions for drawing lithography. Maltose was molded into microneedles while in the glassy state, which provided the structural integrity required for skin penetration. When the active compound is liquid, it can be blended with maltose [50]. The viscosity of maltose was measured using a rheometer (TA Instruments, Rheolyst AR1000L Rheometer, New Castle, DE, USA) equipped with a 4 cm flat plate probe, a 120 µm gap, and a shear rate of 5 s^−1^. Due to its enzymatic degradation by maltase glucoamylase, maltose readily dissolves and has been extensively utilized as a safe biopolymer for encapsulating bioactive compounds [51].

Wu, Y. et al. developed dissolving bilayer microneedles to deliver proteins to the back of the eye for retinal disorders. Using polymers like PVA/PVP, they optimized microneedles to penetrate the sclera and dissolve rapidly while maintaining protein bioactivity. The microneedles were non-irritants and showed enhanced protein permeation through the sclera compared to patches, establishing an efficient and safe intraocular protein delivery system [52]. A nanosuspension of cholecalciferol was prepared using PVA and PVP as stabilizers for enhanced transdermal delivery. The nanosuspension was embedded into hydrophilic polymer-based dissolving microneedles. These dissolving microneedles prepared with PVA/PVP blends showed good mechanical properties and efficient skin penetration [53].

#### 2.1.4. Coated Microneedles

An adaptable delivery method is a coated microneedle. A single microneedle patch can deliver diverse substances, encompassing small molecules, deoxyribonucleic acid (DNA), proteins, viruses, and microparticles. Studies have shown that coated microneedles can provide DNA and proteins into the skin with minimal invasiveness [54,55]. The primary objectives of the study were to establish even coatings on microneedles and to identify the range of particles and molecules suitable for coating onto microneedles. Initially, microneedles were crafted individually or in clusters from stainless steel sheets. Subsequently, a novel micron-scale dip-coating method using a generally recognized as safe (GRAS) coating formulation was developed to apply even coatings on individual microneedles and arrays consistently [56]. Compounds like bovine serum albumin, calcein, vitamin B, and plasmid DNA were all coated using this method [57]. Microparticles with a diameter of 1 to 20 μm and modified *vaccinia virus* were also coated. Coatings were selectively applied to the needle shafts, designed to dissolve in the skin of a cadaverous porcine within 20 s. Histological analysis confirmed that microneedle coatings were injected during insertion and remained intact without wiping off [56]. The research presents an immediate, adaptable, and manageable method for coating microneedles with proteins, DNA, viruses, and microparticles, enabling rapid delivery into the skin [55]. Various materials are used for coated microneedles, including stainless steel, titanium, polycarbonate, silicon, and polymer blends. Stainless steel microneedles offer excellent mechanical strength and durability, making them suitable for clinical applications [58]. Titanium microneedles are lightweight, biocompatible, and corrosion-resistant, ideal for biomedical applications [59]. Silicon microneedles offer precise control over geometry and dimensions, biocompatibility, and compatibility with microfabrication techniques [60]. Polymer blends, such as PEG and PVA blends, offer tunable mechanical properties and biodegradability, catering to specific requirements for drug delivery, including controlled release and biocompatibility. These materials present a diverse array of options for coated microneedles, enabling researchers to select the most suitable material based on the application’s requirements and biocompatibility considerations [61].

D. Jakka et al. investigated the development of polymer-coated polymeric (PCP) microneedles for the controlled release of APIs in dermal and intravitreal drug delivery. PCP microneedles demonstrated sustained release of lidocaine hydrochloride for up to 9 h in skin tissue and voriconazole intravitreally for 6 h, suggesting their potential for controlled drug delivery [62].

#### 2.1.5. Coating Single Microneedles

Single microneedles were dip-coated by being placed horizontally within a droplet of coating solution held in a 200 μL large-orifice pipette tip. Each microneedle was then immersed in 20–30 μL of the coating solution. Both the microneedle and the pipette tip were securely clamped horizontally on a manual linear micropositioner (A1506K1-S1.5 Unislide, Velmex, Bloomfield, NY, USA) positioned opposite each other. The microneedle was manually maneuvered and observed through a stereo microscope (SZX12, Olympus, Center Valley, PA, USA), facilitating insertion and removal from the liquid droplet [54].

#### 2.1.6. Hydrogel-Forming Microneedles

The newest variety of microneedles, HFMs, were first noted in 2012. HFMs are made of crosslinked hydrogels, which can swell and have a more diverse mechanism of action than other materials. Hydrogel-based flexible matrices (HFMs) exhibit swelling upon skin insertion owing to their inherent hydrophilicity, facilitating water absorption. This characteristic makes them suitable for biomedical purposes such as interstitial fluid (ISF) uptake, predominantly within the dermal layer of the skin, encompassing cellular environments in tissue interstices [63]. HFMs are regarded as minimally invasive because, due to their microscale nature, they do not interact with or activate pain receptors positioned deeper in the dermis layer of the skin. Moreover, hydrogel-forming microneedles (HFMs) address certain limitations of traditional microneedles. Specifically, HFMs offer a variable drug release rate and increased loading capacity. These characteristics are often linked to the polymer crosslinking ratio, a parameter challenging to control in traditional microneedles [64,65]. The feasibility of achieving sustained transdermal delivery of high-dose metformin HCl through a hydrogel-forming microneedle patch has been explored. This approach holds promise for mitigating specific gastrointestinal side effects and addressing fluctuations in negligible intestine absorption associated with oral administration [66]. The microneedle layer, which was made from an aqueous mixture of 20% weight-percent poly (methyl-vinyl ether-co-maleic acid) and 7.5% weight-percent poly (ethylene glycol), was crosslinked by esterification and used to assemble patches (two layers) [67]. More than 90% of metformin from homogeneous drug reservoirs with a molecular weight of 10,000 Da was successfully retrieved. The drug reservoir dissolved in less than 10 min in phosphate-buffered saline (PBS) with a pH of 7.4. The microneedle achieved consistent penetration of Parafilm^®^ M, a validated skin model. In vitro experiments conducted in a controlled laboratory setting confirmed that the microneedle effectively improved the permeation of metformin HCl through neonatal porcine skin samples obtained from the dermatome [68].

#### 2.1.7. Biodegradable Microneedles

Biodegradable microneedles provide an innovative approach to drug delivery, enabling precise and targeted administration of therapeutics using minimally invasive, disintegrating structures [69]. Utilizing biodegradable polymers such as PLA, chitosan, PGA, or PLGA, biodegradable microneedles can serve as a more patient-friendly option to traditional sustained-delivery techniques. After application, these microneedles break down in the skin, enabling months of continuous medication release. However, to fully utilize the degradation property, they must be inserted and present on the skin for several days [70].

Qiu, Li et al. used biodegradable polymer microneedles made of PLA to enhance drug permeability in skin. They found 600 μm high microneedles were mechanically stable, and 800 μm deep with 256 microneedles per cm^2^ were most effective. Drug concentration increased drug permeation amount, while higher viscosity decreased it. Prolonged drug administration stabilized permeation. In vivo, these microneedles effectively delivered insulin, reducing blood glucose levels in diabetic mice [71]. Scientists created biodegradable microneedles with multiple layers to regulate drug release. They used a sequential spraying process with PLGA and PVP. Tests confirmed strong layer adhesion and successful skin penetration with biphasic drug release. They examined a model protein drug’s integrity within the microneedles, finding minor structural changes. In vitro release studies showed controlled kinetics, with a blank PLGA layer reducing initial burst release. Confocal microscopy verified the barrier formation. Overall, the study highlights these microneedles’ potential for transdermal drug delivery [72].

**Table 1 pharmaceutics-16-01398-t001:** Types of microneedles.

Sr. No.	Type of Microneedles	Material Used	Fabrication Method	Reference
**1**	Solid microneedles	(i) Silicon microneedles (ii) Metal microneedles (iii) Polymer microneedles (iv) Ceramic microneedles	Etching	[73,74]
**2**	Coated microneedles	(i) Stainless steel (ii) Glass (iii) Chitosan	Spraying	[75,76]
**3**	Dissolving microneedles	(i) Polymers (ii) Sugars (iii) Proteins	Encapsulation	[20,77,78,79]
**4**	Hollow microneedles	(i) Metals (ii) Silicon (iii) Glass (iv) Polymers (v) Nickel	Centrifugation	[77,80,81,82]
**5**	Hydrogel-forming microneedles	(i) PVP (ii) Hydrophilic polymers	Dispersion of solution	[83,84]
**6**	Biodegradable microneedle	(i) PVP (ii) PLGA (iii) PGA	Molding or casting	[72,85]

**Table 2 pharmaceutics-16-01398-t002:** Microneedle material selection and their characteristics.

Material	Mechanical Characteristics	Biocompatibility	Drug Loading Capacity	Transparency	Advantages	Disadvantages	Applications	Reference
**Silicon**	Excellent mechanical strength	Biocompatible	Moderate to high	Not transparent	Good mechanical properties	Brittle and easily broken	Ocular drug delivery with a tear-soluble contact lens and penetrates into the cornea.	[86]
**Metal**	High mechanical strength	Biocompatible	Moderate to high	Not transparent	High mechanical strength	Corrosion risk, potential allergic reactions	Diagnostics and drug delivery	[59]
**Polymer**	Flexible	Biocompatible	Low to moderate	Not transparent	Flexible and easily fabricated	Limited mechanical strength, potential degradation	Drug administration, biosensing	[87]
**Glass**	Brittle	Biocompatible	Low to moderate	Transparent	Excellent optical transparency	Fragile and can break easily	Delivery of solution and nanoparticles in sclera	[88]
**Dissolving**	Varies	Biocompatible	Low to moderate	Varies	Dissolves entirely in the body	Short needle length, limited drug loading capacity	Drug delivery, localization, and sustained release	[79]
**Hydrogel**	Soft and adaptable	Biocompatible	Low to moderate	Not transparent	Soft and biocompatible	Mechanical weakness, potential swelling	Sustainable ocular drug delivery	[12]
**Ceramic**	High mechanical strength	Biocompatible	Moderate to high	Not transparent	High mechanical strength, good chemical stability	Difficulty in fabrication, brittleness	Drug administration, biosensing	[12,45]
**Biodegradable**	Varies	Biocompatible	Moderate to high	Varies	Dissolves completely in the body	Limited mechanical strength, potential degradation	Drug administration, biosensing	[89,90]

### 2.2. Fabrication Techniques

#### 2.2.1. Photolithography

Photolithography, sometimes called “optical lithography”, utilizes light to imprint patterns from a photomask onto a light-sensitive chemical called a “photoresist”, which is coated onto a substrate. This method involves selectively eliminating unexposed areas. Photolithography follows a top-down methodology, with processing protocols and materials differing based on particular implementations. Nevertheless, they adhere primarily to a standard procedure, illustrated in Figure 4 [91]. Before coating the photoresist, thorough cleaning of the substrate, usually a silicon wafer, is essential to remove contaminants such as solvent stains (like methyl, alcohol, and acetone), atmospheric dust, residues from equipment and operators, microorganisms, aerosol particles, and similar impurities. This procedure necessitates operation within cleanroom facilities featuring a precisely controlled environment to maintain minimal airborne particulates, stable temperature, air pressure, suitable humidity levels, minimal vibration, and controlled lighting conditions [92]. In particular circumstances, notably in biomedical applications, the silicon wafer acts as a solid base for additional material layers. This choice stems from its favorable characteristics: rigidity, flatness, affordability, and smoothness [93].

The silicon wafer is commonly covered with a thin layer of photoreactive materials, typically monomers, oligomers, or polymers. Near-infrared (NIR) light is favored over UV light due to its reduced photo-damaging effects and enhanced penetration depth when patterning biomaterials such as proteins and cells. Depending on the characteristics of the photoresist, diverse radiation ranges can be utilized, encompassing electron beams, ion beams, and X-rays [94]. At the heart of photolithography lies the fundamental concept of inducing chemical changes in the photoresist upon light exposure. UV light is passed through a photomask featuring opaque patterns printed on a transparent substrate. Subsequently, these patterns are transferred onto the photoresist. During the subsequent development stage, the outcome of the remaining photoresist differs depending on whether a positive or negative photoresist is utilized. Positive photoresists dissolve in the exposed areas, while negative photoresists dissolve in the unexposed regions [95].

#### 2.2.2. Micromolding

Micromolding techniques encompass manufacturing methodologies employed to fabricate diminutive components featuring intricate details on the microscale, typically ranging from micrometers to millimeters. These methodologies are pivotal in sectors including microelectronics, biotechnology, medical devices, and aerospace, where exacting and miniature parts are indispensable [96]. These techniques facilitate the formation of intricate geometries, high aspect ratios, and precise tolerances in miniature components, offering benefits such as enhanced performance, minimized material consumption, and cost reduction. They frequently entail molding processes tailored for microscale applications, including micro-injection molding, micro-compression molding, micro-casting, micro-electroforming, micro-hot embossing, micro-powder injection molding, and micro-transfer molding, explained in more detail in Table 3 [97]. By harnessing these methodologies, manufacturers can address the escalating demand for miniaturized products spanning various sectors, thereby propelling technological advancements and fostering innovation.

#### 2.2.3. 3D Printing

Microneedles are pivotal in diverse biomedical applications, including drug delivery and biosensing. Advanced 3D printing methodologies facilitate their precise fabrication and customization [34]. Stereolithography (SLA) and digital light processing (DLP) employ light to cure photosensitive resins in a layered fashion, yielding microneedles characterized by sharp tips and intricate features. Fused deposition modeling (FDM) utilizes the extrusion of thermoplastic filaments, although its capability to achieve extremely minute features may be limited, as presented in Figure 5 [108].

Two-photon polymerization (TPP) is founded on the principle of photopolymerization, where a focused laser beam selectively cures a liquid resin [109]. Direct ink writing (DIW) enables the controlled extrusion of viscous bioinks or materials onto substrates, which is ideal for crafting biodegradable microneedles tailored for drug delivery applications. Selective laser sintering (SLS) employs laser energy to sinter powdered materials layer by layer, offering adaptability with a range of materials, including those pertinent to biomedical contexts [110]. Each technique possesses distinct advantages and can be personalized to meet specific microneedle requirements such as length, diameter, and material properties. Furthermore, post-processing steps like sterilization and surface modification are commonly employed to refine the performance and enhance the biocompatibility [29] of microneedle fabrication using 3D printing technology. Additional information is provided in Table 4, which includes detailed information on microneedle fabrication using 3D printing technology.

### 2.3. Advantages of Microneedle Drug Delivery

Ophthalmic microneedles provide a targeted and minimally invasive system for delivering drugs directly to the eye’s tissues, bypassing systemic circulation and reducing off-target effects. Their small size and precise insertion minimize discomfort and tissue damage, making them well-suited for delicate ocular structures. Microneedle-based drug delivery systems provide a means of controlled release for therapeutic agents, improving drug bioavailability and extending therapeutic effects, thus contributing to enhanced patient compliance. Their customizable design also allows tailored approaches to specific eye conditions, optimizing treatment outcomes. With the potential for combination therapy and improved stability, ophthalmic microneedles hold promise for revolutionizing the therapy of various eye diseases and disorders. Microneedle-mediated drug delivery presents numerous merits, rendering it an increasingly appealing methodology within the domains of pharmaceuticals and healthcare, as given in Table 5.

Microneedles also have the advantage of a targeted and controlled-release drug delivery system. Controlled DDSs are engineered to precisely administer therapeutic agents to specific cells, tissues, or organs [120]. Augmentations involving hydrogels, nanoparticles, or siRNA encapsulation within liposomes enhance their efficacy [121]. In contrast to conventional DDSs with limitations such as systemic application and constrained delivery efficiency, microneedle-based controlled transdermal DDSs emerge as a solution [122]. Microneedle-based systems significantly improve the efficiency and precision of drug delivery, offering benefits such as targeted localization, decreased dosing frequency, and simplified self-administration. Consequently, they foster enhanced patient compliance. This technological innovation is especially advantageous for individuals with specific health conditions, including young children, the elderly, and those experiencing challenges such as vomiting and nausea [123]. In polymeric microneedle systems, drug release occurs when drug molecules move from the inner polymeric matrix to its outer surface and are released into the surrounding tissue. The regulation of drug-release kinetics is essential for achieving controlled drug delivery [120]. Chen et al. enhanced a long-acting microneedle patch for blood glucose control by optimizing its rapid separation feature. The hydrogel microneedle system replicates normal insulin secretion, offering a prompt response to elevated glucose levels and controlled release, thereby improving postprandial blood glucose control [124]. The adoption of microneedle drug delivery technology has been linked to a decrease in side effects when compared to traditional methods. This is attributed to microneedle delivery’s targeted and controlled nature, allowing for precise administration of therapeutic agents. The minimally invasive approach of microneedles reduces the potential for adverse reactions, as they primarily target specific cells, tissues, or organs [125]. Furthermore, the controlled release and localized action of drugs through microneedles contribute to a more favorable pharmacokinetic profile, minimizing systemic exposure and thus reducing the likelihood of systemic side effects. Consequently, microneedle drug delivery technology shows promise in improving the overall safety profile of therapeutic interventions [126]. Migdadi et al. researched hydrogel-forming microneedles aimed at transdermal delivery of metformin to alleviate gastrointestinal side effects commonly associated with oral administration. Their findings underscored enhanced permeation and bioavailability of the drug, facilitated by the microneedles developed in their study [127].

### 2.4. Case Studies of Drug-Loaded Microneedles

The case studies presented encompass various drug-loaded microneedle formulations, each demonstrating unique characteristics and encapsulation efficiencies. These formulations utilize various nanoparticle systems, nanosuspensions, solid lipid nanoparticles (SLNs), colloidal nanoparticles, nano-microparticles, inclusion complexes with cyclodextrins, microcrystal particles/powder, micelles, and solid dispersions, as given in Table 6. Solid lipid nanoparticles (SLNs) and nanosuspensions appear frequently among the formulations, indicating their popularity and effectiveness in ocular drug delivery. For instance, paclitaxel-loaded SLNs exhibited an encapsulation efficiency of 54.13 µg per patch, showcasing the potential of SLNs for sustained drug release [128]. Similarly, capsaicin-loaded colloidal nanoparticles demonstrated an impressive encapsulation efficiency of 99.9%, highlighting their suitability for efficient drug distribution to the eye [129]. Additionally, nanosuspensions, such as those containing methotrexate and TA, demonstrated promising characteristics with encapsulation efficiencies ranging from 2.48 mg to 92.52 µg, indicating their potential for delivering an extensive range of drug doses [130]. Moreover, inclusion complexes with cyclodextrins, such as those of levonorgestrel and TA, exhibited encapsulation efficiencies of 66.94 µg to 92.52 µg, suggesting their ability to enhance drug solubility and stability [131,132]. Other formulations, such as solid dispersions and matrix interactions, also showed promising results. For instance, atorvastatin calcium trihydrate in solid dispersion form exhibited encapsulation efficiencies ranging from 1.9 to 3.4 mg, indicating its potential for delivering relatively higher drug doses [133]. Likewise, lidocaine hydrochloride formulated via matrix interaction demonstrated an encapsulation efficiency of 3.43 ± 0.12 mg, indicating its potential for sustained drug release and prolonged therapeutic effect [134]. Furthermore, using PLGA nanoparticles (NPs) for drug delivery was highlighted in several case studies, demonstrating their versatility and efficacy in ocular drug delivery. For instance, PLGA NPs loaded with OVA exhibited encapsulation efficiencies ranging from 4.15 µg to 10 µg, indicating their potential for delivering antigens for ocular immunotherapy [33,135]. Overall, the diverse range of formulations and their respective encapsulation efficiencies showcased in these case studies underscores the potential of various nanoparticle systems for efficient and targeted ocular drug delivery, paving the way for improved treatment outcomes in ophthalmology. However, further research is acceptable to optimize these formulations for clinical translation and address scalability and regulatory approval challenges.

### 2.5. Evaluation Parameters for Ocular Microneedles

Ocular microneedles represent a groundbreaking advancement in ophthalmology, enabling precise delivery of therapeutic compounds to the eye. These micron-sized needles penetrate ocular barriers with minimal invasiveness, promising improved efficacy and patient comfort [11]. However, ensuring microneedle systems’ safety, efficacy, and reliability requires a thorough evaluation process. The evaluation parameters for ophthalmic microneedles are outlined as follows: A critical evaluation parameter for ocular microneedles is biocompatibility. Materials used in microneedle fabrication must be non-toxic and non-irritating to ocular tissues. Biocompatibility assessments involve in vitro studies to evaluate cell viability, proliferation, and inflammatory response, alongside in vivo studies to assess tissue compatibility and immune reactions [151,152].

Mechanical strength is pivotal for the effective penetration and drug delivery of ocular microneedles. Evaluation involves testing microneedles’ durability and fracture resistance under various conditions, including insertion forces, repeated use, and storage conditions. These assessments ensure reliable performance during administration and mitigate the risk of needle breakage or deformation [37,153]. Insertion efficiency: Efficient penetration of ocular barriers significantly impacts drug delivery efficacy. Evaluation of insertion efficiency includes assessing penetration depth, insertion force, and reproducibility of needle insertion. Methods like optical coherence tomography (OCT) and confocal microscopy offer real-time visualization and measurement of microneedle penetration, offering valuable visions into the efficacy of drug delivery to specific ocular tissues [154,155].

Drug loading and release: Efficient loading and controlled release of therapeutic agents is imperative for ocular microneedle efficacy. Evaluation parameters encompass drug loading capacity, release kinetics, and stability of loaded drugs within microneedles. In vitro release studies simulate ocular conditions to determine drug release profiles over time, ensuring precise and sustained delivery to the target site [156]. Pharmacokinetics and pharmacodynamics: Comprehensive evaluation involves pharmacokinetic and pharmacodynamic studies to assess drug distribution, absorption, and therapeutic response. Microdialysis, ocular imaging, and pharmacological assays provide valuable data on drug bioavailability, tissue distribution, and pharmacological effects, guiding microneedle design and formulation optimization [12]. Safety and tolerability: Evaluation extends to safety and tolerability assessments to ensure minimal adverse effects and patient comfort. Studies evaluate ocular irritation, inflammation, tissue damage, and visual disturbances associated with microneedle administration. Biocompatibility, sterility, and pyrogenicity testing further confirm the safety profile of ocular microneedle systems for clinical use [157].

### 2.6. Biocompatibility and Safety Considerations

Designing a long-acting drug delivery microneedle must consider several factors to ensure effective and efficient medication delivery. Critical considerations for their biocompatibility and safety include using biocompatible and non-toxic materials, such as metals, polymers, and biodegradable substances, which ensure minimal inflammatory responses and non-toxic degradation products [61]. Mechanical properties are critical; the microneedles must be strong enough to penetrate ocular tissues without breaking and appropriately sized and sharp to minimize tissue damage and pain [43]. Sterilization and maintaining aseptic conditions during manufacturing are essential to prevent infections. Microneedles must also promote rapid healing of the insertion site and deliver drugs in a controlled and targeted manner [158]. To prove their safety and efficacy, extensive preclinical research in animal models and human clinical trials is needed. Regulatory compliance, including meeting FDA or EMA standards and post-market surveillance, is crucial for successful implementation in clinical settings [159]. Here are a few crucial design considerations:

#### 2.6.1. Needle Length and Geometry

In-depth research has been conducted on microneedle array-based transdermal DDSs to determine their biocompatibility and viability as a commercially viable method of transporting small and large molecules (peptides, drugs, and proteins). Microneedles, manufactured with remarkable precision owing to advancements in microfabrication technology, have demonstrated exceptional efficacy in transdermal delivery by puncturing the stratum corneum. Generally, these microneedles range in length from 150 to 1500 μm, with widths spanning 50 to 250 μm and diameters between 1 and 25 μm. By puncturing the skin, microneedles create micron-sized pores, and these channels serve as a straight path for drug delivery [160]. Patient compliance and pain management are vital for the success of Minnesota-based drug delivery. However, the length and quantity of microneedles were found to be essential for pain management. The 400-microneedle patch was painless, each microneedle measuring 150 μm in length. However, the pain score escalated significantly by seven- and threefold, respectively, when the needle length was extended from 500 to 1500 μm (while maintaining a constant number of needles) and when the number of microneedles was increased by tenfold (while maintaining a continuous length of 620 μm) [161].

Microneedle patches are formed by arranging microneedles in arrays on the backing of a patch. However, for microneedles to serve as effective drug delivery systems, they must meet specific criteria. Microneedles are available in various sizes and shapes, with needle-shaped geometries (sharp, tapered, conical, or bevel-tipped), microblades, or blunt projections being the most common. Regarding array design, fabrication techniques for microneedle arrays typically yield “in-plane” or “out-of-plane” systems. “In-plane” arrays are oriented perpendicular to the surface, whereas “out-of-plane” arrays are aligned parallel to the surface. Davis et al. were pioneering investigators who examined the effect of microneedle geometry on insertion force using both in vitro and in silico experimental approaches. Their findings revealed a direct association between increasing microneedle cross-sectional area and insertion force [162]. On the contrary, the study found a consistent elevation in fracture forces concerning microneedle wall thickness, wall angle, and tip radius variations. Therefore, the researchers determined that the fracture forces corresponding to various geometries were greater than the insertion forces [163]. After analyzing various geometries and dimensions of in-plane silicon microneedle designs, it was found that the skin’s inherent resistance to puncture significantly affects the insertion force, thereby influencing the microneedle’s penetration capability [93,95]. The sharpness of microneedles is a critical factor influencing tissue damage and the subsequent healing process. Microneedle sharpness is typically quantified by the tip radius of curvature, with smaller radii indicating sharper needles. Sharp microneedles, which often have a tip radius ranging from a few nanometers to a few micrometers, penetrate the skin more efficiently and with less force than blunter needles. This reduced penetration force minimizes mechanical damage to the surrounding tissue, leading to smaller, more precise incisions that limit tissue trauma. Minimized tissue damage from sharp microneedles translates into faster and less complicated healing; reduced tissue trauma results in lower levels of inflammation and a quicker re-epithelialization process. Additionally, the risk of infection is diminished, as more minor wounds are less susceptible to bacterial invasion [164,165]. To avoid breakage and buckling during use, microneedles must have adequate strength and stiffness, influenced by their material properties, geometric design, and shaft width, which generally falls between 10 μm and 300 μm [166]. Gill et al. tested microneedles of varying lengths, widths, thicknesses, and tip angles to determine their pain levels compared to a 26-gauge hypodermic needle. They found that all microneedles caused significantly less pain, with length having the most substantial impact; longer microneedles caused more pain. Increasing the number of microneedles also increased pain, but tip angle, thickness, and width did not significantly affect pain levels. Shorter and fewer microneedles were less painful, supporting their potential for less painful transdermal drug delivery [167].

#### 2.6.2. Material Biocompatibility

Microneedles facilitate the administration of diverse medications, covering small molecules, peptides, vaccines, proteins, and nucleic acids. The suitability of a particular drug for microneedle delivery hinges on various factors:

**Physicochemical properties of the drug:** Drugs possessing suitable physicochemical properties, such as low molecular weight, adequate solubility, and stability, are generally more compatible with microneedle applications [156]. Particle size is important; the drug particles should be small enough to integrate uniformly into the microneedle matrix without causing blockages or structural issues, with nanoparticles often preferred for enhanced solubility and absorption [168]. Enhancing the solubility of poorly soluble drugs is crucial for overcoming the challenge of delivering small doses via microneedles. Increasing the solubility means higher doses of these drugs can be effectively incorporated into the small dimensions of microneedles. Techniques to enhance solubility include using prodrugs, surfactants, liposomes, salt preparation, pH adjustment, and nanoparticle control technology [156].

**Formulation considerations:** The formulation of the drug plays a crucial role in its compatibility with microneedles. Different formulation approaches, such as nanoparticle encapsulation, microspheres, or hydrogels, can enhance drug compatibility [169]. Lahiji et al. investigated the effects of various microneedle manufacturing parameters, including manufacturing and storage temperatures and drying conditions. They found that maintaining low temperatures during manufacture, using mild drying conditions, ensuring appropriate polymer concentration, and incorporating a protein stabilizer could preserve lysozyme activity at 99.8 ± 3.8%. The study underscores the significance of optimizing manufacturing conditions to maintain protein activity [170].

**Stability:** Drugs must remain stable during microneedle fabrication and storage. Some drugs may require special protection or stabilization techniques to prevent degradation [156]. Permeation enhancers: In some cases, chemical permeation enhancers may be necessary to facilitate drug delivery across the skin barrier. Drug compatibility may vary based on the microneedles’ design and fabrication method. Experimental investigations and formulation refinements are frequently required to ascertain the compatibility of a specific drug with microneedles [169]. Selecting materials and formulations to preserve protein drug stability is challenging, particularly in large-scale storage and production for clinical applications. Chen et al. developed a microneedle incorporating phenylboronic acid, demonstrating glucose responsiveness and temperature stability for insulin delivery in diabetes treatment [171]. Antibody delivery encounters multiple challenges, including reduced efficacy and the risk of immunogenicity resulting from protein inactivation. To address these issues, it is crucial to ensure the stability of the antibody within the microneedle and carefully consider formulation aspects. Zhu et al. examined the stability of vaccine-loaded microneedles. They discovered that using trehalose during manufacturing provided significantly greater stability than sucrose, retaining 80% of the initial antigenicity under stress conditions (60 °C for 3 months) [172].

**Loading capacity:** The microneedles’ loading capacity indicates the drug volume that can be accommodated within the microneedle array [173]. Several factors influence the loading capacity: the microneedles’ size, geometry, and material impact their loading capacity. Microneedles can vary in length, width, and shape, allowing for different drug-loading possibilities [165]. Various drug formulation strategies can be employed to improve loading capacity. For example, drugs can be layered onto the surface of microneedles, encapsulated within the microneedles (such as in dissolving microneedles), or incorporated into biodegradable matrices that surround the microneedles [166]. The required therapeutic dose of the drug also influences the loading capacity. Microneedles are typically used for delivering small to moderate drug doses, especially for localized or targeted applications [174]. The loading capacity of microneedles is often limited compared to other DDSs like patches or injections. However, researchers continuously optimize microneedle designs and drug formulations to increase loading capacity and improve drug delivery efficiency [156]. This study identifies biocompatibility and minimal invasiveness as essential for advancing next-generation microneedle medical treatments. Consequently, selecting candidate materials should prioritize biocompatibility and low cell toxicity. Traditional materials employed in medical applications often consist of metals to ensure robustness and rigidity. Thus, non-ferrous metals emerge as promising candidates for microneedles. The materials must withstand insertion while remaining intact during drug release as given in Table 7. Microneedle fabrication frequently employs silicon, biodegradable polymers, and metals such as stainless steel or titanium [175].

## 3. Route of Administration for Ocular Microneedles

Ocular microneedles are innovative in administering medications directly into the eye’s tissues. While their primary application involves intrastromal injection into the corneal stroma, researchers actively investigate diverse methods to refine drug delivery and address specific ocular conditions. The ophthalmic medication routes are presented in Figure 6. Below are several alternative routes for the administration of ocular microneedles:

### 3.1. Intrastromal Injection

Intrastromal injection delivery of ophthalmic microneedles involves the precise insertion of ultra-thin needles directly into the stroma layer of the cornea. These microneedles are designed to penetrate the corneal tissue with minimal trauma, allowing for targeted delivery of medications or therapeutic agents. Once inserted, the microneedles can release drugs into the stroma, bypassing barriers such as the tear film and corneal epithelium, thereby enhancing drug bioavailability at the target site while minimalizing systemic side effects. This approach holds promise for treating various ophthalmic conditions more effectively and with reduced patient discomfort compared to conventional methods [176].

### 3.2. Intravitreal Injection

Intravitreal injection delivery of ophthalmic microneedles involves precisely inserting extremely fine needles directly into the eye’s vitreous cavity. These microneedles are designed to penetrate the ocular tissues with minimal trauma, facilitating the targeted distribution of medications or therapeutic agents into the vitreous humor. Once inserted, the microneedles can release drugs directly into the vitreous, allowing for enhanced drug bioavailability at the site of action while minimizing systemic side effects [177].

### 3.3. Subconjunctival Injection

Subconjunctival injection delivery of ophthalmic microneedles involves the precise insertion of tiny needles just beneath the conjunctiva, the thin membrane covering the white part of the eye. These microneedles are intended to penetrate the conjunctival tissue with minimal discomfort, enabling targeted delivery of medications or therapeutic agents to the underlying ocular structures. Once inserted, the microneedles can release drugs directly into the subconjunctival space, allowing for localized treatment of many eye conditions such as inflammation, infection, or glaucoma. This approach provides the benefit of prolonged drug release and minimized systemic side effects compared to traditional topical eye drops [178].

### 3.4. Suprachoroidal Injection

Suprachoroidal injection delivery of ophthalmic microneedles involves the precise insertion of tiny needles into the space between the sclera and choroid, the outer layers of the eye. These microneedles are designed to penetrate this space with minimal trauma, allowing for targeted delivery of medications or therapeutic agents to the choroid and adjacent tissues. Once inserted, the microneedles can release drugs directly into the suprachoroidal space, enabling localized treatment of various ocular conditions such as macular edema, choroidal neovascularization, or uveitis [179].

### 3.5. Transscleral Delivery

Transscleral delivery of ophthalmic microneedles involves the insertion of tiny needles through the sclera, the tough outer layer of the eye, to deliver medication or therapeutic agents to the intraocular tissues. The microneedles are crafted to penetrate the sclera with minimal tissue damage, facilitating accurate drug delivery to the posterior eye area and covering the retina and choroid. Once inserted, the microneedles can release drugs directly into the sclera, from where they can diffuse into the intraocular tissues, providing localized treatment for macular degeneration, diabetic retinopathy, or glaucoma. This approach offers the advantage of bypassing ocular barriers and achieving high drug concentrations at the target site, potentially improving treatment efficacy while minimizing systemic side effects [180].

## 4. Therapeutic Agents Delivered via Microneedles

Microneedles have gained popularity for ocular therapy due to their ability to deliver medications for various ocular diseases. These medications include anti-inflammatory agents, anti-VEGF agents, and antiglaucoma agents. The type of microneedle used depends on the treatment [11]. Dissolving microneedles are suitable for anterior segment diseases, as they can be applied similarly to contact lenses, improving patient acceptability. Hollow and solid microneedles can target diseases that affect the posterior segment of the eye, requiring precise administration protocols in clinical settings. The appropriateness of the microneedle type for a specific application can be validated through the delivery of model drugs [181]. Thakur et al. investigated administering small molecules and macromolecules to the posterior eye segment by dissolving PVP microneedles. These microneedles exhibited robustness and sharpness, enabling successful penetration through corneal and scleral barriers. This penetration caused a tenfold improvement in the delivery of macromolecules compared to conventional topical methods [37]. The study further explored the application of dissolving microneedles for delivering PLGA-encapsulated ovalbumin (OVA) nanoparticles into the sclera, facilitating prolonged release. A bilayered microneedle design was employed, focusing therapeutic molecules solely within the needle segment to augment drug bioavailability. Successful insertion of FITC-OVA nanoparticle-loaded microneedles into the sclera was achieved. Microneedles significantly improved the delivery efficiency of macromolecules and nanoparticles to the posterior segment of the eye, leading to increased therapeutic effectiveness for retinal diseases [35].

### 4.1. Antibiotics

The skin, the body’s largest organ, harbors pathogenic bacteria, contributing to skin and soft tissue infections (SSTIs) affecting 7–10% of hospitalized individuals. SSTIs pose health, cosmetic, and economic challenges [182]. Typical systemic antibiotic treatments may foster resistance owing to inadequate concentrations at the site of infection and exposure to healthy microbiota [183]. Microneedles have emerged as potential antibiotic delivery platforms for intra- and transdermal applications. Microneedle arrays of dissolvable polymers offer a minimally painful and easily applicable solution, enabling high local drug concentrations. This approach seeks to address the limitations of systemic antibiotic administration in dermatology [184]. Dissolvable microneedle arrays have proven to be an efficient means of delivering a range of antibiotics across or within the skin, such as gentamicin (GEN) [185], chloramphenicol [186], tetracycline [187], cephalexin [188], doxycycline [189], polymyxin [190], vancomycin (VAN) [191], and clindamycin [192]. Ziesmer et al. devised hybrid microneedle arrays with a dual-layer configuration—an external water-soluble layer containing VAN and an internal water-insoluble layer with plasmonic nanoparticles for photothermal effects. These arrays exhibited significant drug loading, attained temperature elevations of up to 60 °C via NIR irradiation, and demonstrated synergistic suppression of methicillin-resistant Staphylococcus aureus (MRSA) proliferation. This preliminary investigation highlights the potential effectiveness of these arrays as an innovative treatment approach for MRSA-related skin infections [193]. Vázquez et al. created dissolving polymeric microneedle arrays to administer GEN transdermally in low-resource settings. The arrays demonstrated mechanical resilience and efficient penetration in skin simulants. In vitro experiments confirmed the successful delivery of GEN, while in an animal model, diverse doses yielded dose-dependent plasma levels. This method holds promise for in vivo transdermal antibiotic delivery, mitigating the necessity for trained healthcare personnel, dose computations, and proper injection equipment in resource-limited environments [185]. Turner et al. fabricated economical hydrogel microneedles through 3D printing for transdermal delivery of amoxicillin and VAN. These microneedles exhibited effective drug delivery, enhanced resolution, and mechanical strength, successfully penetrating skin grafts with minimal damage. The distinctive drug-loading method obviated the necessity for an external reservoir, enabling controlled antibiotic release. The hydrogel microneedles displayed robust antimicrobial properties, suggesting their potential for minimally invasive transdermal antibiotic administration [194]. In 2017, Bhatnagar et al. employed dissolving microneedles composed of PVP/PVA, consisting of a 6 × 6 needle array, to administer the antibiotic besifloxacin directly to bacterial infections in the cornea. These besifloxacin-loaded microneedles were engineered to penetrate the corneal barrier effectively, resulting in better management of ocular infections with higher besifloxacin concentrations in corneal tissue compared to conventional drug solutions. Furthermore, unlike free besifloxacin solution, the microneedles exhibited depot-like characteristics within the cornea, prolonging the therapeutic effect, reducing the need for frequent topical drug application, and ultimately enhancing patient compliance [195]. Albadr et al. similarly documented the development of rapid-dissolving microneedles loaded with amphotericin B for treating intracorneal infections. The formulation aimed at incorporating amphotericin B into the fast-dissolving matrix employed a blend of PVP and hyaluronic acid. Analysis using multiphoton microscopy unveiled the establishment of an amphotericin B reservoir after intra-scleral administration of the microneedles. Direct incorporation of amphotericin B resulted in enhanced drug loading and bolstered mechanical strength, as evidenced by the authors [79].

### 4.2. Steroids

Microneedles offer a promising avenue for addressing various skin conditions, including psoriasis, dermatitis, eczema, acne, and skin cancer [77]. Commonly used topical corticosteroids have vasoconstrictive, immunosuppressive, anti-inflammatory, and anti-proliferative properties [196]. However, traditional formulations may reduce patient compliance due to odor, greasy texture, frequent dosing, stickiness, and potential side effects. Microneedles offer a minimally invasive and site-specific delivery approach, addressing these concerns and presenting a promising alternative for treating inflammatory skin diseases [197]. Dawud et al. propose a novel drug delivery system for treating inflammatory skin diseases utilizing microneedles loaded with dexamethasone (DEX)-loaded nanoparticles (NPs). These PLGA NPs ensure controlled drug release. The microneedles, incorporating DEX-NPs, exhibit enhanced skin insertion and mechanical strength. Dissolution studies reveal that NP-loaded microneedles dissolve within 15 s, releasing NPs into the skin. This system aims to surpass traditional topical treatments’ constraints by offering self-administration, improved patient adherence, and regulated drug release, thereby enhancing therapeutic efficacy [198]. Jang et al. engineered dissolving microneedles comprising the therapeutic molecule triamcinolone acetonide (TA) to improve minimally invasive transdermal drug delivery for conditions like atopic dermatitis. They addressed TA’s poor solubility by introducing a suspension and creating high-dose TA-dissolving microneedles through sonication and polymer optimization. In vitro and in vivo testing showcased its potential as an effective and high-dose treatment for skin inflammatory conditions requiring substantial steroid doses [199]. The study investigates using a biodegradable microneedle patch to augment the efficacy of topical steroids in treating prurigo nodularis. In vitro and clinical studies revealed enhanced steroid penetration and improved treatment outcomes when the microneedle patch followed the application of topical steroids. The results suggest that this approach could be beneficial for managing challenging skin conditions such as prurigo nodularis [200].

### 4.3. Anti-VEGF Agents

Abnormal neovascular diseases, including AMD, diabetic retinopathy, and CNV, significantly contribute to irreversible blindness. In these cases, the notable feature is the overexpression of VEGF, a protein that fosters the growth of fragile new blood capillaries and heightens the permeability of existing ones. The resulting overexpression leads to blockages, leakage, and bleeding, contributing to the progression of sight-threatening diseases [201]. Anti-VEGF-based therapies are widely acknowledged as the primary strategy for inhibiting neovascularization and protecting against retinal vascular disorders. Biomacromolecules like ranibizumab, aflibercept, bevacizumab, and pegaptanib are commonly utilized anti-VEGF agents. Thus, it is critical to develop dissolving microneedles to effectively administer protein drugs to the posterior segment of the eye [202]. Kim et al. showcased the effectiveness of coated microneedles in delivering bevacizumab to the corneal stroma, successfully suppressing neovascularization in rabbits. This has been achieved with significantly lower doses than subconjunctival injection and topical eye drops. The minimally invasive approach demonstrated promising results without observable adverse effects on corneal transparency or structure [176]. Coyne et al. developed polymer microneedles for the localized delivery of DNA aptamers targeting VEGF to treat disorders brought on by overexpression of specific proteins. These microneedles dissolved upon contact with a physiological solution, releasing a concentrated dose of anti-VEGF aptamer. The aptamer-loaded microneedles demonstrated potential as a therapeutic tool by diminishing VEGF-mediated endothelial cell tube formation in a tissue phantom [203].

### 4.4. Anti-Inflammatory Agents

Ophthalmic anti-inflammatory drugs are predominantly classified into corticosteroids and nonsteroidal anti-inflammatory agents. Corticosteroids, exemplified by DEX, TA, and fluocinolone acetonide, are widely recognized as the cornerstone therapy for ocular inflammation due to their robust anti-inflammatory effects and potential anti-angiogenic properties. Nevertheless, significant apprehension arises from the adverse effects of corticosteroids, notably increased intraocular pressure (IOP) and the formation of cataracts [204]. Shields et al. showed that one month of use of corticosteroid eye drops significantly increased IOP in some patients. Monitoring and interventions may be necessary. Intravitreal corticosteroid injections for posterior ocular inflammation can cause adverse effects, including elevated intraocular pressure. Microneedles provide a less invasive and targeted method for safely administering anti-inflammatory agents by bypassing ocular barriers [205].

## 5. Applications of Microneedles in Ocular Disease

### 5.1. Age-Related Macular Degeneration

Macular degeneration, scientifically termed AMD, is a pathological condition characterized by potential visual impairment, including blurred or absent central vision [206,207]. The initial stages of the condition typically manifest without apparent symptoms [208]. Subsequently, individuals may progressively deteriorate vision, impacting one or both eyes. Although macular degeneration does not culminate in total blindness, losing central vision poses challenges in recognizing faces, driving, reading, and engaging in routine daily activities. Furthermore, individuals may experience visual hallucinations as part of the clinical presentation [209].

Macular degeneration typically manifests in the elderly population and arises from injury to the macula of the retina. Contributing issues include genetic predisposition and smoking habits. Diagnosis involves a comprehensive eye examination, with severity classification ranging from early to intermediate, and late types further categorized into “dry” and “wet” forms [210]. The prevalence of the dry form accounts for 90% of cases [211]. The differences between dry and wet forms of age-related macular degeneration (AMD) stem from changes occurring in the macula. In dry-form AMD, individuals typically develop drusen, deposits of cellular debris within the macula. These drusen cause gradual damage to the light-sensitive cells, leading to progressive vision loss. Conversely, wet-form AMD involves the growth of abnormal blood vessels beneath the macula, which leads to the leakage of blood and fluid into the retina [212]. The disruption in the balance between the creation of damaged cellular components and their degradation results in the accumulation of harmful products, such as intracellular lipofuscin and extracellular drusen. Early stages of AMD are characterized by incipient atrophy, which manifests as regions of thinning or depigmentation in the retinal pigment epithelium (RPE) preceding the onset of geographic atrophy [213]. In the advanced stages of age-related macular degeneration (AMD), the breakdown of the retinal pigment epithelium (RPE), termed geographic atrophy, and the development of abnormal blood vessel growth, known as neovascularization, are significant factors leading to the loss of photoreceptors and subsequent impairment of central vision [214]. When drusen builds up between the retina and the choroid in the dry (nonexudative) form of AMD, retina atrophy and scarring are caused [215]. On the other hand, in the more severe wet (exudative) variety, choroid-derived blood vessels (neovascularization) sprout behind the retina, causing bleeding and exudate fluid leakage [216].

Initial research revealed that drusen contained numerous immunological mediators. Notably, complement factor H (CFH) plays a crucial role in suppressing this inflammatory process; AMD is significantly associated with a mutation in the CFH gene linked to the disease [217]. As a result, inflammation in the macula and persistent low-grade complement activation have been linked to a pathophysiological model of AMD. The discovery of genetic variations linked to disease in complement component 3 (C3) and other complement cascade components supports this concept [218]. Maintaining a healthy diet, quitting smoking, and getting regular exercise have all been linked to a possible lower incidence of macular degeneration [219]. It is significant to highlight that there is currently no medication or cure for this ailment that would restore vision that has already been lost. Treatment options for wet-form instances include intraocular injections of anti-VEGF medicine or, less frequently, photodynamic therapy or laser coagulation, which may slow the progression of the disease. Dietary supplements may help delay the disease’s development in those who have already been diagnosed with macular degeneration, even if dietary antioxidant vitamins, carotenoids, and minerals do not appear to affect the disease’s beginning [220].

#### Microneedle-Based Therapies for AMD

Despite successful intravitreal injection delivery, approximately 45% of individuals with AMD exhibit unresponsiveness to VEGF drugs. Furthermore, age-related changes in the dynamics of vitreous humor can obstruct the distribution of formulations within the eye, creating hurdles for effective delivery to the posterior segment [221]. Age-related liquification of vitreous humor is associated with complications like macular holes, vitreous detachment, and hemorrhage. The limitations extend to poor permeation of high molecular weight anti-VEGF drugs with short half-lives binding to the blood–retinal barrier and extracellular matrix. These factors collectively hinder the success of posterior segment disease treatment, diminishing predictability and reproducibility in intravitreal pharmacokinetics [222]. Continuing investigations are directed towards exploring microneedle-based treatments for AMD, a progressive ocular ailment impacting the macula and resulting in central vision impairment [223].

Traditional treatment modalities involve invasive intraocular injections, necessitating frequent clinical interventions. Microneedles’ diminutive needle-like structures offer a potential alternative by providing targeted and less invasive drug delivery to specific layers of the eye [224]. The advantages include enhanced patient comfort, improved precision in drug delivery to the affected regions, and the prospect of reducing treatment frequency, as presented in Figure 7. Nevertheless, challenges such as optimizing microneedle design, ensuring safety, and addressing long-term usage concerns persist [225].

Ongoing preclinical trials and investigations are focused on evaluating the safety and efficacy of microneedle-based drug delivery systems for treating AMD. Continuous research is required to refine the technology and validate its clinical utility. For the latest developments, referring to the recent scientific literature, clinical trial databases, and authoritative medical sources is recommended due to the dynamic nature of advancements in this field [226]. Amer et al. engineered microneedle arrays based on polyvinyl alcohol (PVA) hydrogel to deliver immunoglobulin G1 (IgG1), a model protein similar to bevacizumab. Bevacizumab is a monoclonal antibody employed in the treatment of AMD. The production process commenced with creating a master mold using light processing-based 3D printing. Subsequently, the mold’s shape was replicated in an elastomer (Sylgard^®^ 184,Dow, Midland, Michigan, USA ), which was then used to fabricate the final microneedles via molding. In vitro assessments were conducted using a Parafilm/polyethylene/nylon surrogate membrane and a fluid-simulating vitreous humor. These analyses revealed a prolonged release of the active substance, contrasting with the rapid release observed post-injection. The authors emphasized that the arrays of microneedles demonstrated a significantly more consistent drug release pattern than individual injections [227]. Kadonosono et al. explored the application of microneedles for administering tissue plasminogen activator (tPA) and air as a therapeutic approach for sub-macular hemorrhage associated with secondary AMD. Sub-macular hemorrhage is defined by the buildup of blood between the retinal pigment epithelium and the retina. This condition often results from CNV or wet AMD [129,130]. In a phase 1 clinical trial, Canton et al. assessed the safety and tolerability of a singular microneedle delivery of bevacizumab into the SCS utilizing Clearside Biomedical’s exclusive microneedle technology. Four adult patients diagnosed with CNV associated with wet AMD underwent the procedure. Preliminary results indicated successful drug delivery without unforeseen adverse effects. Despite the report of a moderate pain level during administration, no severe adverse events (AEs) linked to bevacizumab or the injection method were recorded. IOP remained stable, and no additional therapeutic interventions were required within the initial two months post-treatment. These outcomes propose the feasibility of safe bevacizumab administration into the SCS using Clearside Biomedical’s microneedle and solely employing topical anesthesia [228]. The current treatment approaches for diseases affecting both the anterior and posterior eye segments demonstrate notable limitations, highlighting the necessity for pioneering strategies [11]. To address the constraints inherent in current ocular drug delivery methodologies, investigators are actively discovering diverse strategies designed to augment the bioavailability of ocular drugs [229]. These strategies involve sophisticated drug delivery systems like nanoparticles, liposomes, and hydrogels. Additionally, researchers are exploring the utilization of prodrugs, permeation enhancers, and device-based methods such as iontophoresis and microelectromechanical systems (MEMSs) [230]. These strategies are devised to boost the penetration and retention of drugs within ocular tissues. Concurrently, the implementation of targeted DDSs holds promise for optimizing drug pharmacokinetics [231]. This method entails delivering therapeutic substances directly to targeted cells within the eye, such as the retina or iris, circumventing the need for diffusion and permeation processes after administration [232].

### 5.2. Diabetic Retinopathy

Diabetic retinopathy is the primary cause of newly detected vision impairment among adults aged 20–74. Within the initial two decades following diagnosis, nearly all individuals with type 1 diabetes and over 60% of those with type 2 diabetes exhibit indications of retinopathy [233]. The Wisconsin Epidemiologic Study of Diabetic Retinopathy (WESDR) revealed that 3.6% of individuals with early-onset diabetes (type 1) and 1.6% of those with late-onset diabetes (type 2) were classified as legally blind. Within patients with type 1 diabetes, 86% of instances of legal blindness were linked to diabetic retinopathy. In contrast, among patients with type 2 diabetes, where other ocular conditions were more prevalent, diabetic retinopathy contributed to one-third of legal blindness cases [234]. Diabetic retinopathy advances from mild nonproliferative stages, marked by increased vascular permeability, to moderate and severe nonproliferative diabetic retinopathy (NPDR), characterized by vascular closure. Subsequently, it may progress to proliferative diabetic retinopathy (PDR), featuring the emergence of new blood vessels on the retina and posterior vitreous surface. Macular edema, characterized by retinal thickening due to leaky blood vessels, can develop throughout these stages. Factors such as pregnancy, puberty, blood glucose control, hypertension, and cataract surgery may accelerate these pathological changes [235]. Vision-threatening retinopathy is rare among individuals with type 1 diabetes during the first 3–5 years following diagnosis or before reaching puberty. However, within the following two decades, nearly all individuals with type 1 diabetes developed some form of retinopathy [236]. At the onset of type 2 diabetes, up to 21% of individuals display signs of retinopathy, with a majority developing varying degrees of retinopathy as time progresses. Impaired vision stemming from diabetic retinopathy can be ascribed to multiple mechanisms. Central vision may be affected by macular edema or capillary nonperfusion [237]. In the realm of proliferative diabetic retinopathy (PDR), the advent of novel vascular formations and the contraction of associated fibrous tissue can deform the retina, leading to tractional detachment of the retina and profound, often permanent vision impairment. Furthermore, these newly formed blood vessels may hemorrhage, exacerbating complications such as retinal or vitreous bleeding. Neovascular glaucoma correlated with PDR can also aggravate visual impairment. The duration of diabetes serves as a critical prognostic factor for both the onset and progression of retinopathy [235]. In the Wisconsin Epidemiologic Study of Diabetic Retinopathy (WESDR), researchers found that among individuals with younger-onset diabetes, the prevalence of any retinopathy increased steadily over time. The prevalence stood at 8% at three years, increasing to 25% at five years, 60% at ten years, and 80% at fifteen years. Initially absent at three years, proliferative diabetic retinopathy (PDR) prevalence rose to 25% by the fifteenth year. Additionally, the incidence of retinopathy rose with a longer duration of diabetes. In those with younger-onset diabetes, the 4-year incidence of developing PDR rose from 0% within the initial five years to 27.9% during years 13–14 of diabetes. Following this, the incidence of developing PDR remained relatively stable beyond the 15-year mark [225,226]. They developed a comprehensive approach for managing diabetic retinopathy depending on the condition’s severity and stage. Critical interventions include vigilant monitoring of blood glucose, blood pressure, and cholesterol, coupled with lifestyle adjustments such as regular exercise, healthy diet, and smoking cessation [238]. For diabetic macular edema, intravitreal injections of anti-VEGF drugs are employed to decrease swelling and prevent vision loss—laser therapies, including focal and pan-retinal photocoagulation, address leaking blood vessels and abnormal retinal vasculature [239]. Surgical procedures like vitrectomy may be necessary for significant bleeding or retinal detachment. Intraocular steroids and anti-VEGF implants offer sustained effects to reduce inflammation and macular edema [240]. Laser retinal photocoagulation targets specific areas to inhibit abnormal blood vessel growth. Regular eye exams are crucial for early detection and timely intervention. Treatment selection depends on individual factors, requiring thorough discussions with healthcare professionals. Early identification and proactive management are essential for preventing vision loss in diabetic retinopathy [241]. Abnormal neovascular conditions leading to irreversible blindness, such as diabetic retinopathy, are driven by increased VEGF. Anti-VEGF agents effectively address angiogenic pathologies by suppressing VEGF-A’s action on blood vessel receptors [201]. However, systemic circulation of excess anti-VEGF agents poses cardiovascular risks. To mitigate these risks, localized and targeted microneedles administration is optimal, allowing for precise, minimally invasive delivery to the eye, reducing systemic side effects, and improving patient compliance [242]. Clearside Biomedical’s suprachoroidal injection of TA (CLS-TA) showed promising results in the PEACHTREE trial. Nearly 47% of patients receiving CLS-TA significantly improved vision, while safety endpoints were met with a lower rate of adverse ocular events (51%) compared to the sham group (58%). The most common adverse effect in the sham group was cystoid macular edema, while CLS-TA patients reported pain (12.5%) and increased IOP (11.5%) [243].

### 5.3. Glaucoma

Glaucoma represents a multifaceted and progressive optic neuropathy characterized by structural impairment to the optic nerve head, leading to corresponding visual field defects. Elevated IOP frequently accompanies this ocular disorder, contributing to mechanical stress and damage to optic nerve fibers. The development of glaucoma involves a complex interaction of genetic, vascular, and environmental factors. Elevated intraocular pressure (IOP) is the primary cause, resulting from an imbalance between the production and drainage of aqueous humor in the eye. High IOP leads to compression and ischemic damage to the optic nerve head, triggering events that ultimately lead to the degeneration of retinal ganglion cells and their axons. This neurodegenerative process is clinically recognizable through the field optic nerve head’s distinctive cupping and the visual field’s gradual decline. Glaucoma can be broadly categorized into open-angle and angle-closure subtypes based on the configuration of the anterior chamber angle, with open-angle glaucoma being the most common form. Primary and secondary glaucoma are also discerned, with the former arising independently and resulting from other ocular or systemic conditions. Timely detection and effective management are imperative in preserving visual function and averting irreversible vision loss in glaucoma. Treatment approaches encompass the use of topical ocular hypotensive medications, laser therapy, and surgical interventions aimed at reducing IOP. Systematic monitoring of optic nerve head morphology and visual field condition is essential for managing individuals with glaucoma. Despite continuous research endeavors, glaucoma persists as a substantial global public health challenge, necessitating ongoing exploration of innovative therapeutic strategies and preventative measures.

#### Microneedle Therapy for Glaucoma Management

Innovative therapeutic approaches augment traditional glaucoma treatments, encompassing the utilization of microneedles explicitly administered to the supraciliary area to enhance the targeted and efficient delivery of pharmaceutical agents, as presented in Figure 8. Investigation into various drugs and administration routes for glaucoma remains an active area of research, signifying ongoing efforts to advance therapeutic modalities in this domain [25]. Jiang et al. presented a study elucidating the application of coated stainless steel microneedles for targeted drug administration in the anterior eye segment, specifically addressing glaucoma treatment. Pilocarpine served as the chosen therapeutic agent, administered via the intra-scleral route, resulting in a substantial increase in drug absorption, approximately 45-fold [76] beyond conventional methods. Additionally, their study demonstrated a 60-fold enhancement in the delivery efficiency of fluorescein compared to traditional topical delivery. The microneedle’s dimensions ranged from 500 to 750 μm [30]. Additionally, the study demonstrated the utility of hollow microneedles for delivering sulforhodamine through the intra-scleral route. Fabricating these hollow microneedles involved using borosilicate micropipette tubes, enabling precise drug delivery at a specific concentration, 10–35 μL from each microneedle within the array. This innovative methodology underscores the potential of microneedle technology in optimizing targeted drug delivery in ocular therapeutics [28].

Patel et al. examined the utilization of hollow microneedles for delivering drugs to the SCS. Their research demonstrated the controlled release of medications embedded in nanoparticle and microparticle formulations within the posterior segment of the eye [88]. Kim et al. aimed to assess the effectiveness of precisely delivering antiglaucoma drugs to the supraciliary space using a hollow microneedle, investigating the potential reduction in required dosage compared to conventional eye drops. Employing rabbits as the experimental model, the findings revealed that the supraciliary administration of drugs resulted in a substantial, dose-dependent decrease in IOP, indicating an approximately 100-fold dose-sparing effect when contrasted with topical administration. This targeted delivery approach showcased promise in enhancing safety and sustaining therapeutic effects over the long term, thereby supporting the notion of achieving such outcomes with a singular injection [180]. Prausnitz et al. investigated the transscleral delivery of antiglaucoma drugs, sulprostone and brimonidine, using hollow microneedles. They inserted a single 33 G stainless steel microneedle, measuring 700 to 800 µm in length, into the sclera to assess its efficacy in delivering the drugs to the supraciliary space. The findings demonstrated that hollow microneedles efficiently and accurately delivered sulprostone and brimonidine to the ciliary body, achieving comparable therapeutic effects with a 100-fold reduction in dosage compared to conventional topical application. This dose-sparing effect of hollow microneedles indicates a potential for significantly reducing adverse effects at non-target sites, thereby improving patient acceptance of this treatment method [180]. Roy et al. devised dissolving microneedles, resembling contact lenses, using a blend of PVA and PVP for the transcorneal delivery of pilocarpine. Ex vivo assessments conducted on excised porcine eyeballs demonstrated that these microneedles, compared to traditional topical drops, facilitated rapid administration, resulting in a significant increase in the flux and availability of pilocarpine in the aqueous humor [244]. Additionally, Khandan et al. emphasized the significant dose-saving advantage of microneedles in delivering pilocarpine for glaucoma relief [245].

### 5.4. Other Ophthalmic Conditions

#### 5.4.1. Retinal Vascular Occlusion

Retinal vascular occlusion is a pathological condition affecting the retina, marked by vision loss. The retina’s optimal function requires constant elimination of blood, oxygen, nutrients, and waste. The retinal vascular system is composed of numerous arteries and veins. If these vessels become blocked or clots form within them, it results in a condition referred to as occlusion. Kadonosono et al. illustrated the application of microneedles for retinal endovascular cannulation as a therapeutic method for central retinal vein occlusion. Historically, glass cannulas have been employed to treat this condition; however, due to frangibility and associated complications, the authors introduced a novel approach employing microneedles to dilate the retinal vein occlusion. The study created a stainless steel microneedle measuring 50 μm in diameter with an inner diameter of 20 μm. The microneedle was attached to a syringe filled with 10 μL of saline solution and engineered to penetrate the occluded central retinal vein for thrombus removal. The study involved the enrolment of 12 patients afflicted with this condition. The outcomes indicated minimal to no complications, and the surgical intervention utilizing micro needles demonstrated efficacy, improving visual acuity [246].

#### 5.4.2. Uveitis

Uveitis encompasses various inflammatory and infectious eye conditions and is a significant factor in vision impairment. The uvea, which includes the iris, choroid, ciliary body, and related structures such as the retina, sclera, and optic nerve, constitutes the eye’s inner lining [212,213]. Uveitis is divided into four main types: anterior, intermediate, posterior, and panuveitis. Anterior uveitis primarily impacts the anterior chamber, intermediate uveitis involves the vitreous, posterior uveitis targets the retina or choroid, and panuveitis affects all three sites—the vitreous, anterior chamber, and retina or choroid [247]. Infectious uveitis commonly results from pathogens such as toxoplasmosis, leptospirosis, onchocerciasis, and cysticercosis. Non-infectious uveitis, on the other hand, is associated with conditions like heterochromic iridocyclitis and chorioretinopathy. Given its diverse etiologies and potential severity, uveitis poses a significant threat to ocular health and underscores the importance of accurate diagnosis and targeted management strategies [248]. Gilger et al. demonstrated microneedles to treat acute posterior uveitis with TA. To overcome challenges in delivering medication to the posterior eye segment, they devised a method utilizing hollow microneedles in the suprachoroidal space (SCS) between the choroid and sclera adjacent to the posterior eye segment. Using a porcine model, the study evaluated the effects of TA in acute posterior uveitis. Results indicated that a 2 mg dose of TA effectively reduced posterior inflammation for up to 3 days, with no observed signs of retinal toxicity or increased intraocular pressure (IOP) following SCS injection [249].

#### 5.4.3. Retinitis Pigmentosa

Retinitis pigmentosa (RP) is a hereditary ocular disease involving damage to rod and cone cells, initially affecting rods and progressing to cones, resulting in night blindness and peripheral vision impairment. RP is characterized by changes in photoreceptor cell pigmentation, presenting as bone spicule formations. The disease is generally categorized into syndromic and non-syndromic forms. Non-syndromic RP results from gene mutations and is further classified into autosomal dominant, autosomal recessive, and X-linked RP based on gene types [218,219]. Many therapeutic strategies exist for managing retinitis pigmentosa, including gene therapy, anti-apoptotic agents, neurotrophic factors, and retinal prostheses. Gene therapy, in particular, shows promise for ocular diseases due to its reduced systemic side effects and demonstrated effectiveness in preclinical and clinical investigations. This method employs viral vectors such as adeno-associated virus (AAV), adenovirus, and lentivirus, along with non-viral vectors including liposomes, compact nanoparticles, various polymers, and polypeptides [250]. Viral vectors are often delivered through the subretinal route to ensure effective transduction. On the other hand, non-viral vectors are introduced via the intravitreal route, albeit with less efficient transduction than viral vectors. Recent advancements in non-viral treatments involve using coated polylactide nanoparticles, featuring a coating formulation comprising human albumin serum and hyaluronic acid [251].

#### 5.4.4. Conjunctivitis

Conjunctivitis, or pink eye, is a common inflammatory condition affecting the thin membrane covering the eye’s white part and inner eyelids. It can result from various factors, including infections, allergies, or irritants, causing symptoms like redness, itching, tearing, discharge, and blurred vision [252]. While mild cases often resolve without treatment, severe or recurrent cases may require medical intervention, typically through topical eye drops or ointments. However, traditional approaches to drug delivery, like eye drops, face challenges such as low bioavailability and poor patient compliance [253]. Ophthalmic microneedles have appeared as an encouraging method for drug delivery in ocular diseases like conjunctivitis. These tiny structures penetrate the ocular surface, delivering drugs directly to the affected tissues. By bypassing ocular barriers and achieving localized delivery, microneedles offer benefits such as decreased systemic side effects, enhanced drug bioavailability, and improved patient compliance [25]. By optimizing drug delivery to the inflamed conjunctiva, microneedles could enhance the efficacy of anti-inflammatory or antimicrobial agents, potentially leading to faster symptom resolution and improved outcomes [166]. Bhatnagar et al. developed polymeric microneedle arrays loaded with besifloxacin to enhance drug delivery through the cornea for treating ocular infections. These microneedles, fabricated using polyvinyl alcohol and polyvinyl pyrrolidone, penetrated up to 200 μm into the cornea and dissolved entirely within 5 min. Their utilization notably enhanced the deposition and permeation of besifloxacin across the cornea, showcasing promise as an efficacious therapeutic approach for ocular bacterial infections [195].

#### 5.4.5. Corneal Neovascularization

Corneal neovascularization, involving abnormal blood vessel growth into the cornea, is associated with various ocular pathologies and poses risks to vision integrity. Conventional treatments like corticosteroids, anti-angiogenic agents, and surgeries have limitations in efficacy and administration frequency. Ophthalmic microneedles, composed of biocompatible polymers, offer a promising solution by enabling the precise delivery of therapeutic agents directly into the corneal stroma [254]. This approach enhances drug bioavailability, reduces systemic side effects, and improves patient compliance. Early studies demonstrate the potential of microneedles in inhibiting neovascularization and maintaining corneal transparency, suggesting a transformative strategy for managing this condition pending further clinical validation [255]. Kim et al. aimed to evaluate the effectiveness of coated microneedles for intrastromal delivery of bevacizumab in treating corneal neovascularization. They observed that a solitary application of these microneedles, carrying 4.4 μg of bevacizumab, notably diminished neovascularization compared to untreated eyes. This underscores a promising, minimally invasive, and highly targeted treatment method with significant dose conservation relative to traditional approaches [176].

## 6. Clinical Trials and Regulatory Considerations

This topic focuses on the current status of clinical trials investigating these microneedles and the complex regulatory challenges of their development.

### 6.1. Overview of Ongoing Clinical Trials

Ongoing clinical trials investigate the potential applications of ophthalmic microneedles, showcasing a surge in interest from various pharmaceutical and biotech companies. These trials aim to assess the safety, effectiveness, and feasibility of microneedle-based drug delivery systems (DDSs) for different ocular diseases. Several ongoing trials specifically investigate microneedles’ potential in treating retinal conditions such as age-related macular degeneration (AMD) and diabetic retinopathy, among others. Advancements in the field include novel formulations, design modifications, and innovative delivery strategies, which contribute to ongoing research efforts, as given in Table 8

Clearside Biomedical, Inc. (Alfareta, GE, USA) conducted a phase 3 randomized, masked, controlled clinical trial to evaluate the safety and effectiveness of TA Injectable Suspension (CLS-TA 4 mg/mL) for treating macular edema resulting from non-infectious uveitis [256]. In October 2021, FDA approval was granted for XIPERE^®^, a single stainless steel hollow microneedle designed to deliver TA suspension into the suprachoroidal area for treating diabetic macular edema (DME). The SCS Microinjector^®^ device, as shown in Figure 9, enables precise and localized drug delivery into the suprachoroidal space (SCS) by utilizing the pressure difference within this space to disperse the medication. Additionally, a phase 3 randomized trial (PEACHTREE, NCT02595398) was conducted to assess the suprachoroidal injection of CLS-TA [11]. At week 24, the primary endpoint was defined as achieving a best-corrected visual acuity (BCVA) improvement of at least 15 Early Treatment of Diabetic Retinopathy Study (ETDRS) letters. Results revealed that 46.9% of patients who received a 4 mg CLS-TA suprachoroidal injection on Day 0 and at week 12 met this endpoint, compared to only 15.6% in the sham group. Safety endpoints were also met, with just over half of patients (51%) experiencing adverse ocular events, compared to 58% in the sham group. The most common adverse event in the sham group was cystoid macular edema, with 11.5% experiencing elevated intraocular pressure (IOP) and 12.5% reporting pain [243]. The TA formulation (40 mg/mL) was administered, and changes in SCS thickness were measured shortly after in early Clearside Biomedical trials as HULK (NCT02949024). Optical coherence tomography (OCT) was used to confirm that, even though there was a noticeable rise in thickness (65.1 µm to 75.1 µm) right after treatment, there were no appreciable differences in SCS thickness between treated and untreated eyes until nearly five months later [257]. The potential advantages of combining aflibercept were investigated in a phase 2 trial known as TYBEE (NCT03126786). Patients were divided into two groups: the control group received a single intravitreal injection of aflibercept every four weeks until week 12. In contrast, the treatment group received a combined dose of CLS-TA (40 mg/mL) and aflibercept (2 mg/0.05 mL) on day zero, followed by a single aflibercept injection at week 12. Despite no significant disparity in mean BCVA scores, the combination therapy exhibited a reduction in the overall number of required treatments. Importantly, both groups demonstrated similar benefits, suggesting the possibility of alleviating the clinical burden associated with treating diabetic macular edema (DME) [258]. Clearside Biomedical, Inc. also conducted a phase 1/2 study (NCT01789320) to evaluate the safety and tolerability of a single microinjection of TA (TRIESENCE^®^) within the suprachoroidal space (SCS) of patients with non-infectious uveitis. The research included non-infectious intermediate, posterior, or pan-uveitis patients who underwent a singular suprachoroidal injection. Specifically, the injection involved administering 4 mg of TA in 100 μL directly into the eye under study. Subsequently, a comprehensive follow-up was conducted for 26 weeks to examine the outcomes and effects of this treatment method. In this study, nine subjects were enrolled, with three participants having anterior and intermediate uveitis, one with intermediate uveitis alone, and five with pan-uveitis. Of all the 38 AEs that were reported, the majority had mild to severe severity. Most AEs (almost half) were ocular. Four subjects who complained of ocular pain at or adjacent to the injection site reported the most frequent adverse event. Every systemic adverse event was unrelated to the drug under examination. There were no increases in IOP associated with steroids, and none of the subjects needed medication to decrease their IOP. The visual acuity of all eight efficacy-evaluable patients improved. Through week 26, the average improvement in visual acuity was greater than two lines for four participants who did not require extra therapy. At week 26, macular edema was present in three out of four patients; in two, it had decreased by at least 20%. After these results, they initiated a phase 3 study in patients with ME related to non-infectious uveitis [259]. The University Hospitals Leuven conducted a phase I experiment for central retinal vein occlusion using robot-aided retinal vein cannulation with ocriplasmin infusion. In addition, a robotic system that ensured proper needle alignment and a pump that adjusted the infusion rate managed the microneedle administration. At 26 weeks, macular edema substantially declined in all four patients under study, indicating the viability of ocriplasmin delivery using microneedles. However, there was one case of a microneedle tip breaking, suggesting that there may be more safety issues with the microneedle itself than with the formulation [260].

### 6.2. Regulatory Challenges

#### 6.2.1. FDA and International Approvals

In 2022, Prausnitz and colleagues at Clearside Biomedical introduced XIPERE^®^, the first microneedle product for ocular injections, receiving FDA approval in 2021. Three clinical programs assessed XIPERE^®^: MAGNOLIA, a non-interventional extension study; PEACHTREE, a phase 3 clinical trial; and AZALEA, an open-label safety trial. FDA approval for XIPERE^®^ was based on data from the PEACHTREE trial, involving 160 patients with uveitis-related macular edema. XIPERE^®^ became the inaugural treatment for uveitis macular edema to achieve clinical success with a BCVA primary endpoint. The primary efficacy measure aimed to determine the percentage of patients achieving at least a 15-letter improvement in BCVA following a 24-week follow-up. Notably, the PEACHTREE trial revealed that by week 24, a significantly higher percentage of patients treated with XIPERE^®^ (47%) compared to those in the control arm (16%, *p* < 0.01) experienced at least a 15-letter improvement in BCVA [261].

#### 6.2.2. Safety and Efficacy Requirements

Ensuring the safety of ophthalmic microneedles is critical, requiring a thorough examination of potential AE and implementing risk mitigation strategies during clinical trials. Meeting efficacy requirements involves establishing robust clinical endpoints that convincingly demonstrate the therapeutic benefits of microneedle-based interventions. Challenges related to the selection of appropriate control groups, maintaining blinding procedures, and employing sound statistical methodologies underscore the complexity of clinical trials evaluating ophthalmic microneedles [11]. While microneedles hold promise for enhancing ocular drug delivery, various hurdles require resolution to assist with the effective translation of these technologies from experimental environments to clinical applications. Firstly, it is essential to have a comprehensive knowledge of the biomechanical properties of ocular tissues. Variations in these properties at different locations, such as the limbus versus the equator of the sclera, affect both the force and depth of microneedle penetration and recovery rates. These factors need a thorough evaluation to design optimal microneedles for effective and consistent drug delivery. For example, given the considerable elasticity of the sclera, meticulous attention to microneedle design elements such as needle length, geometry, and interspacing is imperative. Additionally, the biomechanical characteristics of ocular tissues undergo alterations with age and diverse pathologies, necessitating their inclusion in microneedle design considerations. A uniform solution may prove inadequate for diverse ocular microneedle applications [262]. Moreover, for improved application consistency, microneedle administration optimization is essential. Patient comfort and acceptability, especially for self-administration, should be prioritized. Applicator devices ensuring consistent force or injection speed and minimizing procedural steps are desirable to enhance patient acceptability and compliance while avoiding potential injuries from incorrect insertion. Specific challenges exist for different microneedle types. For example, hollow microneedles face difficulties in clinical infusion due to clogging, requiring critical retraction distances. Loading capacity is a consideration to ensure sustained therapeutic levels. Solid microneedles are limited by array surface area, impacting drug doses, while hollow microneedles depend on the drug’s ability to overcome ocular barriers once in the sclera [82]. Accurate dosing poses a challenge, especially for biologics prone to degradation. Careful consideration of fabrication procedures, formulation with stabilizers, and attention to storage, packaging, and transport conditions are necessary to address these issues [263].

## 7. Challenges and Future Directions

### 7.1. Current Limitations of Microneedle-Based Ophthalmics

It is important to understand that despite the minimally invasive nature of the mentioned microneedle systems, there is still a risk of infection and inflammation at the administration site. These complications could lead to additional discomfort and pain. Designing patches with readily removable or dissolvable microneedles is one of the best approaches. Patients will undoubtedly experience discomfort if the patch is left on their eye for an extended period [151]. The hydrophilic polymer-based fastest-dissolving solutions are engineered to reduce discomfort and possible side effects. However, more research is required because there is a shortage of knowledge regarding the hazards of microneedles. Except for single-microneedle injections, no microneedle-based ophthalmic formulations are currently available on the pharmaceutical market. Depending on several variables, including the amount of liquid injected, the patient may find the formulation either painful or bearable. Consequently, each of these elements must be carefully assessed and modified. As previously stated, randomized clinical studies with microneedle arrays are also required. Microneedles can cause discomfort or pain during application if the needles are not sufficiently small or penetrate sensitive eye areas. Even though microneedles are designed to be minimally invasive, any sensation of pain can lead to patient reluctance or fear of the treatment. Minimizing pain perception during application is crucial for patient acceptance and compliance. Microneedle-based ophthalmic therapy may require precise application techniques, which could pose challenges for some patients, especially those with motor impairments or difficulty handling medical devices. Additionally, the need for repeated applications or specialized administration procedures may reduce patient compliance over time. If the treatment regimen is complex or inconvenient, patients may be less likely to adhere to it consistently, potentially impacting treatment efficacy. Intravitreal injections come with established limitations that are widely recognized in the research literature in the realm of ocular treatments. These include the risks of inducing conditions like endophthalmitis, pseudoendophthalmitis, and vitreous detachment. Furthermore, the necessity for repeated injections, typically every month, heightens the potential accumulation of these risks for patients requiring such interventions, and they need to be resolved before being used in ocular applications. Coated microneedles in ocular applications have limitations, such as limited loading capacity, frequent administration, variable drug release rates, and poor repeatability due to coating process reductions in needle sharpness and delivery efficiency. These issues may lead to suboptimal treatment of chronic ocular diseases [76].

#### 7.1.1. Pain Perception and Patient Compliance

Since microneedles are intrusive, using them for ocular therapy may cause pain perception and lower patient compliance. Microneedle implantation in the eye might be uncomfortable, discouraging patients from following their treatment plan. It is necessary to investigate methods to reduce pain and increase patient comfort, such as creating more minor, less invasive microneedles and using a topical anesthetic, to improve patient acceptability and adherence to microneedle-based ophthalmic therapy [232,233]. Size and fabrication challenges: Manufacturing microneedles with diameters appropriate for ocular medication delivery may lead to certain fabrication complications. To prevent harm to the sensitive ocular tissues, microneedles must have the exact size, form, and biocompatibility [264].

#### 7.1.2. Depth Control and Drug Loading and Release

Regulating the depth at which microneedles penetrate ocular tissue is crucial for ensuring efficient drug delivery while minimizing the risk of tissue injury or irritation [43]. Furthermore, optimizing drug loading and release from microneedles is essential to maintain therapeutic concentrations at the target site for the appropriate duration [265].

#### 7.1.3. Biodegradability

To reduce the chance of tissue damage or inflammation and to eliminate the necessity for removal after application, biodegradable materials are favored for microneedles [266].

#### 7.1.4. Patient Acceptance and Compliance

A patient’s acceptance of microneedle-based ocular delivery systems and their adherence to the prescribed treatment plan may be impacted by aspects like comfort, usability, and perceived efficacy [12].

#### 7.1.5. Sterility and Contamination

To avoid infections and other problems, sterility must be maintained during the production, storage, and application of ocular microneedles [267]. Achieving the full potential of ocular microneedles in terms of enhancing patient outcomes and the effectiveness of ocular drug administration will require addressing these constraints through continued research and technological developments. The use of microneedles can cause discomfort or pain during application if the needles are not sufficiently small or penetrate sensitive areas of the eye. Even though microneedles are designed to be minimally invasive, any sensation of pain can lead to patient reluctance or fear of the treatment. Minimizing pain perception during application is crucial for patient acceptance and compliance [268]. Microneedle-based ophthalmic therapy may require precise application techniques, which could pose challenges for some patients, especially those with motor impairments or difficulty handling medical devices. Additionally, the need for repeated applications or specialized administration procedures may reduce patient compliance over time. If the treatment regimen is complex or inconvenient, patients may be less likely to adhere to it consistently, potentially impacting treatment efficacy [269].

#### 7.1.6. Scalability of Manufacturing

Another drawback of microneedle-based ophthalmic therapy is its scalability in manufacturing. Producing microneedles on a big scale with constant quality and accuracy can be difficult and expensive. It might be challenging to scale up current production methods to fulfill the growing demand for microneedles-based ophthalmic medicines. More research is required to develop scalable manufacturing techniques that can produce microneedles at a low cost without sacrificing excellent quality or repeatability [35]. Producing microneedles under aseptic conditions or utilizing terminal sterilization methods is expected to raise expenses substantially. Scaling up microneedle production demands careful deliberation, particularly examining the abundance of small-scale production approaches documented in the literature [268]. Direct micromachining and micromolding are common methods for fabricating micronuclear devices. Production due to high costs and inadequate mass production are barriers to widespread adoption; direct micromachining is used for metal microneedles. Polymer microneedles utilize micromolding techniques, which can present challenges due to the intricate process, involving multiple steps, master molds, and complicated interlocking features at a small scale. Moreover, microneedles can be produced by drawing or SLA-based methods. Since building complicated structures is expensive and these procedures are suitable for mass production, metal microneedles are more commonly fabricated via direct micromachining methodologies, such as micro-milling and laser cutting. Fabricating polymer microneedles is often carried out via micromolding. Although the process of micromolding offers the possibility for large-scale production, there are certain drawbacks, including the requirement for several methods and the creation of master molds from which the microneedle must be removed. Producing additional barbs or interlocking features on a small scale involves intricate manufacturing processes [38].

### 7.2. Future Innovations and Improvements

Incorporating artificial intelligence and machine learning algorithms augments the functionality of these devices, enabling more precise and proactive health monitoring. Nevertheless, several challenges remain to be addressed. One crucial hurdle is optimizing microneedle designs to achieve consistent and reliable penetration without causing tissue damage. Balancing these factors is essential for microneedle-based technologies’ widespread acceptance and effectiveness. There is also a need for advancements in material science to enhance the biocompatibility and stability of microneedles by adopting environment-friendly material to ensure safe and efficient drug delivery over extended periods. Furthermore, developing cost-effective and scalable production methods for microneedles is essential to facilitate large-scale deployment and accessibility across diverse healthcare settings [119].

#### 7.2.1. Potential for Personalized Medicine

Microneedles might be personalized based on a patient’s demography to improve medicine delivery efficiency and shorten healing periods for patient-centric medical care. Advancements in 3D-printed microneedles have enabled the creation of personalized point-of-care diagnostic instruments for tailored medical treatments, fluid sampling from skin tissue, pain-free controlled drug release mechanisms, and bio-signal detection devices [108,270].

#### 7.2.2. Smart Microneedles and Real-Time Monitoring

Wearable microneedle technology emphasizes its potential to monitor vital signs, collect real-time data, and provide customized insights. Incorporating artificial intelligence and machine learning algorithms augments the functionality of these devices, enabling more precise and proactive health monitoring [271]. Combining the synergistic effect of wearable devices with microneedle patches will dominate the diagnosis treatment. This integration facilitates real-time diagnosis through continuous monitoring of biomarkers, while the microneedle patches enable localized and targeted drug delivery based on the collected data [272]. Currently marketed microneedle-enabled patches are presented in Table 9.

## 8. Conclusions

In summary, the realm of ophthalmic microneedle therapy emerges as a promising avenue to deal with the challenges inherent in conventional treatments for ophthalmic diseases and disorders. This review underscores noteworthy progress in microneedle technology, explicitly addressing aspects of design, fabrication, and the diverse DDSs employed. Utilizing microneedle-based DDSs provides distinct advantages, including targeted and controlled release and reducing side effects when juxtaposed with traditional methodologies. The case studies expounded upon in this review accentuate the effectiveness of microneedle therapies in managing distinct ocular conditions such as diabetic retinopathy, AMD, and glaucoma. Furthermore, current clinical trials suggest a rising interest in employing microneedle technology within ophthalmology. However, there are regulatory hurdles that still need to be tackled. Looking forward, the potential for personalized medicine and the development of microneedles with real-time monitoring present exciting opportunities for future advancements in the field. Moreover, the versatility of microneedles in the route of administration, be it solid, coated, or dissolving microneedles—offers flexibility and potential for optimization in delivering therapeutic agents to specific ocular tissues. This multifaceted approach could contribute to improved treatment outcomes and patient adherence. Despite the current limitations, the strides made in microneedle technology offer a glimpse into a more efficient and patient-friendly approach to treating ophthalmic conditions, paving the way for a transformative era in ophthalmic therapy.

## Figures and Tables

**Figure 1 pharmaceutics-16-01398-f001:**
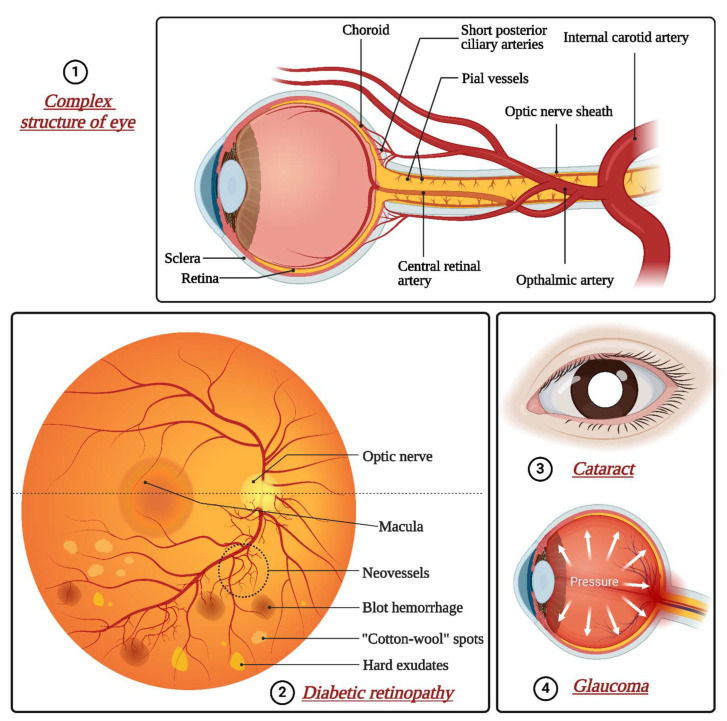
(1) The complex structure of the eye contains three coats to enclose the optically clear aqueous humor, lens, and vitreous body. The outermost coat consists of the cornea and sclera, and the middle coat consists of blood supply to the eye involving the choroid, ciliary body, and iris. (2) Diabetic retinopathy contains “Cotton-wool spots”, which are tiny white areas on the retina, the layer of light-sensing cells lining the back of the eye. (3) Cataracts can develop on aging or injury, resulting in changes in the eye lens involving the breakdown of the protein and fibers to make vision hazy or cloudy. (4) Glaucoma is a chronic, progressive eye disease caused by optic nerve damage leading to visual field loss. An abnormality in the eye drainage causing fluid to build up results in excessive pressure, causing damage to the optic nerve.

**Figure 2 pharmaceutics-16-01398-f002:**
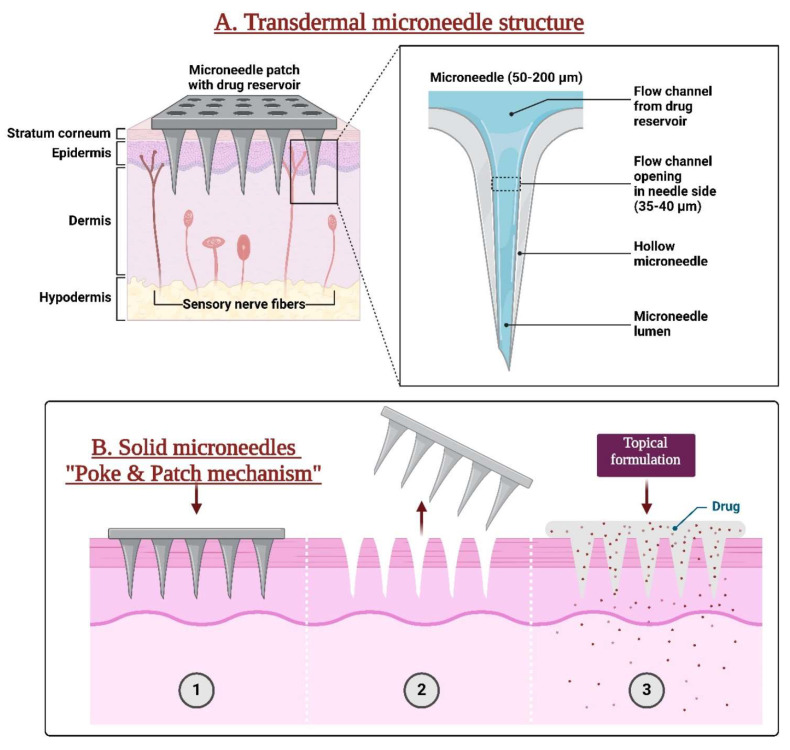
The transdermal microneedle structure and poke and patch mechanism of solid microneedles. The design of the microneedle consists of a flow channel from the drug reservoir, a flow channel opening on the needle side, a hollow microneedle, and a microneedle lumen. The poke and patch mechanism consists of using (1) microneedles to pierce the stratum corneum. (2) The micro-conduits are created in the stratum corneum, (3) a reservoir of the drug applied to the skin and diffuses through the prepared channels to reach the deeper layers of the epidermis.

**Figure 3 pharmaceutics-16-01398-f003:**
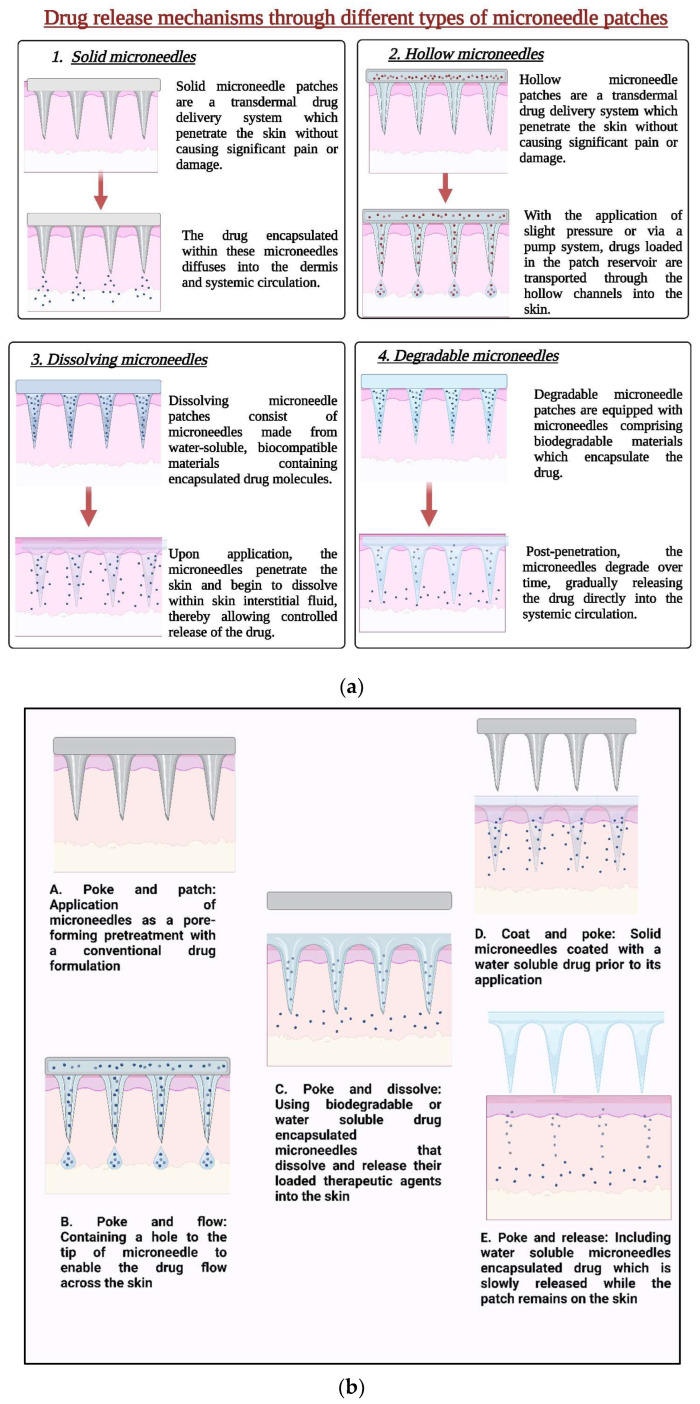
(**a**): The drug release mechanisms through different types of microneedle patches involve the following: 1. Solid microneedle drugs diffuse into the dermis and the systemic circulation. 2. Hollow microneedles involve the application of a slight pressure or pump system to deliver the drug through the hollow channels into the skin. 3. Dissolving microneedles penetrate the skin and begin dissolving into the skin interstitial fluid for drug release. 4. Degradable microneedles degrade over time and allow for gradual drug release directly into the systemic circulation. (**b**): Drug release mechanisms: A. Poke and patch including pore-forming pretreatment with a drug formulation. B. Poke and flow involving the hole into the tip of the microneedle for drug flow across the skin. C. Poke and dissolve involves the dissolution and release of therapeutic agents into the skin. D. Coat and poke involving the solid microneedles coated with a water-soluble drug before application. E. Poke and release includes the release of encapsulated drugs through water-soluble microneedles.

**Figure 4 pharmaceutics-16-01398-f004:**
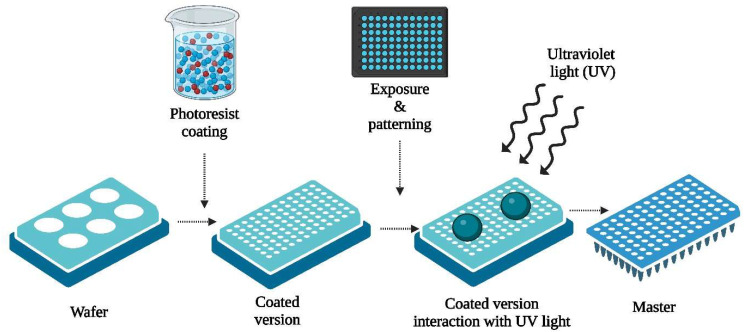
Photolithography: The wafer undergoes a cleaning process to eliminate undesired contaminants. A spin-coating technique is employed to apply the photoresist onto the wafer. The photomask is positioned over the photoresist, and ultraviolet (UV) light is directed through the mask. Solvents remove the unexposed portion of the photoresist, leaving behind the desired patterns.

**Figure 5 pharmaceutics-16-01398-f005:**
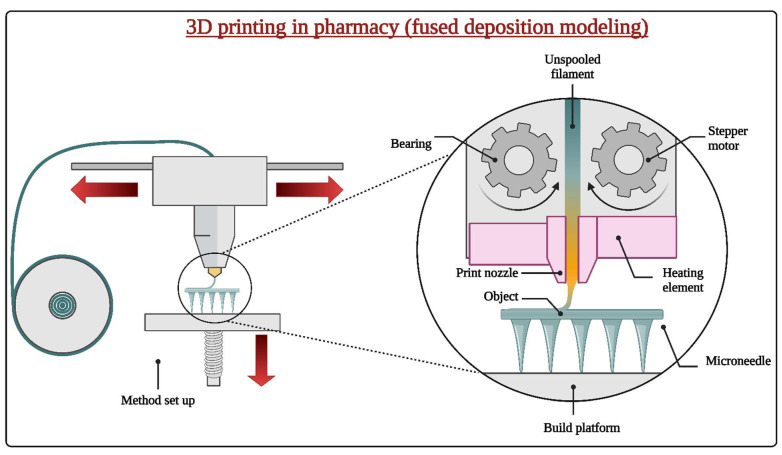
Three-dimensional printing of microneedles by fused deposition modeling.

**Figure 6 pharmaceutics-16-01398-f006:**
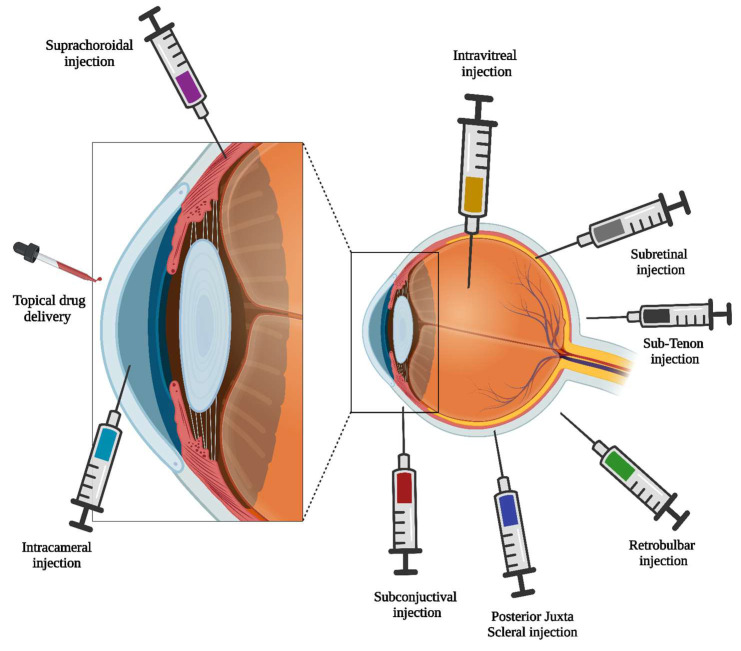
The ophthalmic medication routes involving injection sites for better therapeutics.

**Figure 7 pharmaceutics-16-01398-f007:**
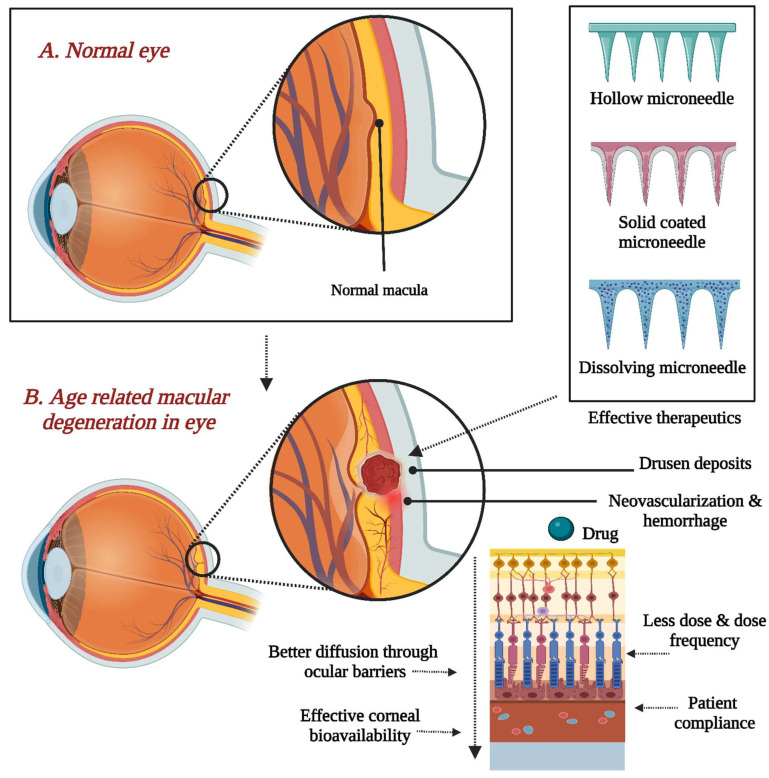
Image of the back of the eye showing intermediate age-related macular degeneration and use of microneedle for the same.

**Figure 8 pharmaceutics-16-01398-f008:**
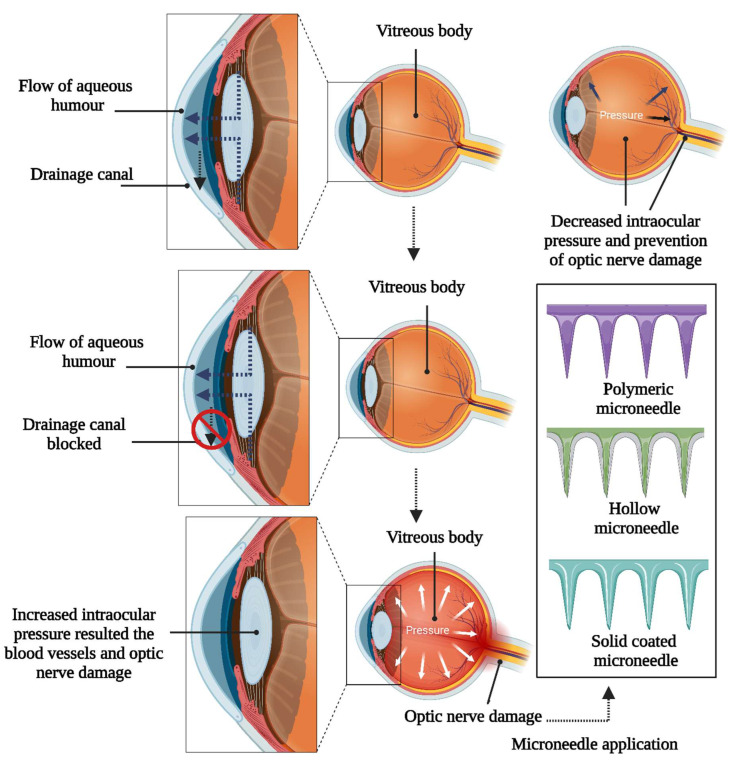
A summary of the onset of glaucoma, which results in increased intraocular pressure and optic nerve damage, along with the use of microneedles for effective glaucoma therapeutics.

**Figure 9 pharmaceutics-16-01398-f009:**
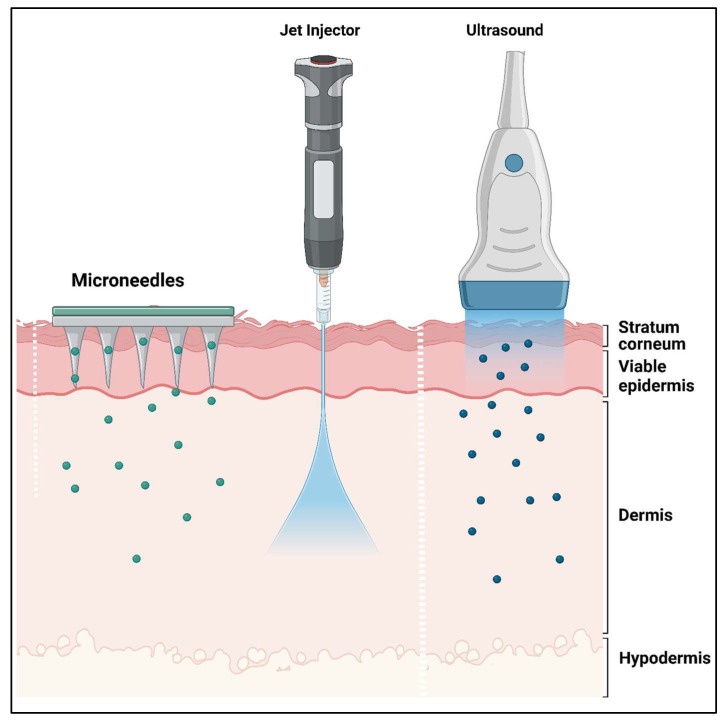
SCS Microinjector^®^ device in comparison with microneedles and ultrasound.

**Table 3 pharmaceutics-16-01398-t003:** Micromolding fabrication techniques.

Technique	Advantages	Disadvantages	Indications	Applications	Ref.
**Micro-Injection Molding**	High precision and repeatability	High initial tooling costs	Mass production of small, intricate parts	Electronics, medical devices, automotive components	[98]
**Micro-Compression Molding**	Suitable for thermosetting polymers	Limited to certain types of materials	Molding small components with precise dimensions	Packaging, aerospace components, microfluidic devices	[99]
**Micro-Extrusion Molding**	Continuous production of micro-sized profiles	Limited to materials with good melt flow properties	Production of microtubing, microfilaments	Medical tubing, micro-cables, microfluidic channels	[100]
**Hot Embossing**	High replication fidelity and resolution	Requires high-precision molds and equipment	Fabrication of microstructures on polymer substrates	Microfluidic devices, optical components, biosensors	[101]
**Micro-transfer molding**	Allows for assembly of pre-formed micro-components	Complex assembly process	Integrating microscale components onto substrates	MEMS fabrication, microelectronics assembly	[102,103]
**Soft lithography**	Versatile for patterning soft materials	Limited to certain types of soft materials	Patterning of elastomers and hydrogels at the microscale	Bioengineering, microfluidics, flexible electronics	[104,105]
**Laser micromachining**	High precision and flexibility in feature creation	Limited to certain materials and geometries	Prototyping, microfabrication of complex structures	Micro-optics, MEMS devices, microfluidics	[76,106]
**Nanoimprint lithography**	High-resolution patterning at nanoscale	Requires specialized equipment and expertise	Nanotechnology, semiconductor manufacturing	Nanophotonics, nanoelectronics, optical devices	[107]

**Table 4 pharmaceutics-16-01398-t004:** Examples of microneedles fabricated by SLA, FDM, and TPP 3D printing.

Fabrication Method	Conjunct Technology	Material(s)	Design	Advantages	Disadvantages	Application(s)	Reference
**TPP**	-	IP-S photoresist	Hollow microneedles	Minimal post-processing	Material limitations	Ocular drug delivery	[81]
**FDM**	Chemical etching	PLA	Cylindrical microneedles	Support material dissolvability; low cost	Warping and shrinkage; Anisotropic mechanical properties	Ocular drug delivery	[12,81]
**SLA**	Micromolding	Resin for master microneedles; carboxymethyl cellulose for microneedles	Conical microneedles	High Resolution	Material Limitations	Ocular drug delivery	[111]

**Table 5 pharmaceutics-16-01398-t005:** Microneedle advantages.

Sr. No.	Advantages	Description	Reference
**1.**	Minimally Invasive	(i) Microneedles are tiny, causing minimal trauma during drug delivery. (ii) Patients experience reduced pain and discomfort compared to traditional injections.	[31]
**2.**	Improved Patient Compliance	(i) Microneedles enhance patient acceptance due to their less invasive nature. (ii) Allows for convenient self-administration, improving patient compliance.	[112]
**3.**	Enhanced Bioavailability	(i) Microneedles enable targeted delivery, improving drug absorption. (ii) Particularly beneficial for drugs with poor oral bioavailability.	[15]
**4.**	Rapid Onset of Action	(i) Facilitates quick drug delivery into the bloodstream, leading to a rapid onset of therapeutic effects.	[113]
**5.**	Preventing First-Pass Metabolism	(i) Bypass the digestive system, preventing first-pass metabolism in the liver.	[114]
**6.**	Improved Stability of Biologics	(i) Enables the delivery of biologics (proteins, peptides) with enhanced stability, preventing degradation.	[115]
**7.**	Tailored Release Profiles	(i) Microneedles can be designed for controlled and sustained drug release, ensuring predictable pharmacokinetics.	[116]
**8.**	Reduced Needlestick Injuries	(i) Smaller needles reduce the risk of needlestick injuries, improving safety.	[117]
**9.**	Potential for Self-Administration	(i) Empower patients to self-administer treatments, reducing healthcare costs and improving convenience.	[118]
**10.**	Versatility	(i) Applicable to various administration routes, including transdermal, intradermal, and mucosal surfaces.	[119]

**Table 6 pharmaceutics-16-01398-t006:** Case studies of drug-loaded microneedles.

Sr. No.	Drug	Potential Applications	Loading Per Patch	Formulation Type	Composition/ Characteristics	Reference
**1.**	Paclitaxel	Treatment for a range of malignancies, including lung, ovarian, and breast cancer	54.13 µg	Solid lipid nanoparticles (SLNs)	Cetyl palmitate and tricaprin, 230 nm	[128]
**2.**	Capsaicin	Topical analgesia for localized pain relief	EE—99.9%	Colloidal nanoparticles	HA and PVP (ratio 1:1), 167 ± 4 nm	[129]
**3.**	Vitamin D3/cholecalciferol	Vitamin D supplementation for individuals with deficiency	265 ± 32 µg	Nano-microparticles	PLGA, 400 nm to 3.6 µm	[136]
**4.**	IR-780	Near-infrared fluorescence imaging for tumor detection	-	SLNs	Cetyl palmitate and tricaprin, 230 nm	[128]
**5.**	Doxycycline	Management of rosacea symptoms	0.84 ± 0.02 mg	SLNs	100 nm	[137]
**6.**	Albendazole	Control of other parasitic infections (e.g., trichinellosis)	0.94 ± 0.03 mg	SLNs	100 nm	[137]
**7.**	Cisplatin	Management of bladder cancer	-	Lipid NPs	DOTAP, cholesterol, and DSPE-PEG-AA	[138]
**8.**	Itraconazole	Therapy for fungal nail infections (onychomycosis)	3.3 mg	Nanosuspension	300 nm	[139]
**9.**	Rilpivirine		4 mg	Nanosuspension		[140]
**10.**	Methotrexate (free acid)	Treatment of rheumatoid arthritis	2.48 mg	Nanosuspension	680 nm	[130]
**11.**	Dutasteride	-	11/12 % (*w*/*w*)	Nanosuspension	-	[141]
**12.**	Curcumin	Treatment of wounds and burns	10.9 ± 1.1 µg	Nanosuspension	520 ± 40 nm	[142]
**13.**	Ivermectin	-	0.86 ± 0.07 mg	Nanosuspension	98.12 ± 7.76 nm	[143]
**14.**	Levonorgestrel	Contraception (long-acting reversible contraception)	66.94 µg	Inclusion complexes with cyclodextrins	Hydroxypropyl-*β*-cyclodextrin (HP-*β*-CD)	[131]
**15.**	TA		80.28 to 92.52 µg	Inclusion complexes with cyclodextrins	(HP-*β*-CD)	[132]
**16.**	Etonogestrel	Contraception (long-acting reversible contraception)	550 µg	Microcrystal particles/Powder	10–30 µm	[144]
**17.**	Lumefantrine	Treatment for simple malaria brought on by strains of Plasmodium vivax and falciparum	8806 ± 461 µg	Nanosuspension	321.00 ± 16.50 nm	[133]
**18.**	Artemether	-	30,027 ± 69.5 µg	Nanosuspension	148.10 ± 4.27 nm	[133]
**19.**	Atorvastatin calcium trihydrate	Management of hypercholesterolemia	1.9 to 3.4 mg	Solid dispersion	-	[133]
**20.**	TA	-	117.06 ± 9.07 µg	Nanosuspension	264 nm	[145]
**21.**	Leuprolide acetate	Hormonal therapy for transgender individuals	14.3 µg	Solid dispersion	-	[146]
**22.**	Shikonin	Promotion of wound healing	0.805 ± 0.017 µg/mg	Micelles	130 ± 8 nm	[147]
**23.**	Finasteride	Treatment of benign prostatic hyperplasia (BPH)	47.36 ± 0.92 µg	Lipid NPs	Glyceryl monostearate and squalene, 180 nm	[148]
**24.**	Lidocaine hydrochloride	Pain management during medical or cosmetic procedures (e.g., injections, tattooing)	3.43 ± 0.12 mg	Matrix interaction	-	[134]
**25.**	Diethylcarbamazine	Treatment of lymphatic filariasis (elephantiasis)	0.55 ± 0.00 mg	SLNs	100 nm	[137]
**26.**	OVA	-	10 µg	PLGA NPs	358 nm	[135]
**27.**	5-aminolevulinic acid	Management of superficial basal cell carcinoma. Therapy for acne vulgaris	69.38 ± 4.89 µg	Matrix interaction	-	[149]
**28.**	Methotrexate	Management of psoriasis	Up to 65.3 ± 2.9 µg	Matrix interaction	-	[150]
**29.**	OVA	Immunization and vaccination against specific antigens or pathogens	4.15 ± 1.93 µg (delivered 24%)	PLGA NPs	170 nm	[33]
**30.**	Lidocaine hydrochloride	Local anesthesia for minor surgical procedures	3.43 ± 0.12 mg	Matrix interaction	-	[134]

**Table 7 pharmaceutics-16-01398-t007:** Metal biocompatibility for medical applications.

Hypersensitivity-inducing element	Cr, Co, V
Poor cellular compatibility element	Cu, Co, V, Fe
Excellent cellular compatibility element	Mo, Ti, Sn, Zr
Enhanced mechanical strength	Zr, Sn
*β*-phase stabilizing element	Ta, Nb, V, Cr, Mo, Fe

**Table 8 pharmaceutics-16-01398-t008:** Ongoing clinical trials on ophthalmic microneedles.

NCT Number	Sponsor	Drug	Phase	Dose	Time	Status	Indication
**NCT02747030**	Universitaire Ziekenhuizen Leuven	Ocriplasmin intravenously	Phase I	-	12 December 2016–11 August 2017	Completed	Central retinal vein occlusion
**NCT03203447**	Clearside Biomedical, Inc.	Suprachoroidal CLS-TA	Phase III	4 mg in 0.1 mL	5 March 2018–18 December 2018	Terminated	Macular edema
**NCT03126786**	Clearside Biomedical, Inc.	IVT aflibercept	Phase II	4 mg in 0.1 mL	11 July 2017–17 April 2018	Completed	Diabetic macular edema
**NCT02949024**	Clearside Biomedical, Inc.	Suprachoroidal CLS-TA	Phase I/II	4 mg in 0.1 mL	10 November 2016–17 October 2017	Completed	Diabetic macular edema
**NCT03097315**	Clearside Biomedical, Inc.	Suprachoroidal CLS-TA	Phase III	4 mg in 0.1 mL	4 April 2017–24 January 2018	Completed	Non-infectious Uveitis
**NCT02595398**	Clearside Biomedical, Inc.	Suprachoroidal CLS-TA	Phase III	4 mg in 0.1 mL	17 November 2015–18 January 2018	Completed	Macular edema with non-infectious uveitis
**NCT02255032**	Clearside Biomedical, Inc.	CLS-TA	Phase II	0.8 mg in 0.1 mL	October 2014–January 2016	Completed	Macular edema with non-infectious uveitis
**NCT02895815**	Janssen Pharmaceutical K.K.	CNTO 2476 (6.0 × 104 cells) in 50 μL	Phase II	-	9 April 2018–19 August 2022	Withdrawn	Visual acuity

https://clinicaltrials.gov (accessed on 1 April 2024).

**Table 9 pharmaceutics-16-01398-t009:** Microneedle-enabled patches on the market.

Product Name	Type of Microneedle	Indication	Reference
Micronject^™^	Solid/dissolving	Drug delivery for various ocular conditions	[273]
TheraJect^™^	Solid/dissolving	Retinal diseases	[26]
SmartPatch^™^	Dissolving	Dry eye syndrome	[274]
Micropoint^™^	Hollow	Drug delivery for macular degeneration	[85]
Visulex^™^	Solid	Intraocular pressure reduction	[275]

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
