# Peer review of "Revolutionizing Eye Care: Exploring the Potential of Microneedle Drug Delivery"

_pharmaceutics, 2024, doi:10.3390/pharmaceutics16111398_

Round 1
Reviewer 1 Report
Comments and Suggestions for Authors
The authors provide an extensive review the literature related to microneedle drug delivery as it pertains to the eye. The authors also review recent relevant clinical trials as well as microneedle devices developed in academic laboratories as well as those available commercially.
The manuscript is very well organized and written and provides a global overview of the field.
Nine authors contributed to the manuscript. In a few places throughout the manuscript, minor changes that can be made to the start of a paragraph. One example is on Line 31 of Page 2 where clarity would benefit from a new paragraph beginning just before the last two words of that line. The authors are encouraged to read through the manuscript with fresh eyes to find and make such minor clarifications.
In line 25 of Page 44, the authors refer to Figure 9, but this reviewer could not find Figure 9 in the PDF file.
The manuscript would benefit from a section discussing the sharpness of microneedles and how that relates to tissue damage and subsequent healing of the tissue. In addition, to the extent that other authors report the tip sharpness (or radius of curvature), the authors are encouraged to include that information in their review.
Comments on the Quality of English Language
The manuscript would benefit from minor editing to enhance clarity.
Author Response
Response to Reviewer Comments
We are incredibly grateful to the expert reviewers for dedicating their time to reviewing our manuscript and providing encouraging feedback thoroughly. Their reviews and suggestions have greatly assisted in enhancing the clarity and quality of this manuscript. We have carefully addressed each comment raised by the reviewers individually. To easily keep track of the changes and corrections, we have highlighted them in yellow in the revised manuscript.
Reviewer #1
Comment 1: One example is on Line 31 of Page 2, where clarity would benefit from a new paragraph beginning just before the last two words of that line.
Response: Thank you for your suggestions. We have revised this paragraph. The new sentence begins with "delivering drugs."
Comment 2: In line 25 of Page 44, the authors refer to Figure 9, but this reviewer could not find Figure 9 in the PDF file.
Response: Thank you for your valuable comment and suggestion. The figure for the SCS Microinjector has been added to the revised manuscript on page 46.
Comment 3: The manuscript would benefit from a section discussing the sharpness of microneedles and how that relates to tissue damage and subsequent healing of the tissue. In addition, to the extent that other authors report the tip sharpness (or radius of curvature), the authors are encouraged to include that information in their review.
Response: We thank the reviewer for the insightful observation. We have included a separate section, 2.6.1, in the revised manuscript on page 29, lines 36 to 54, and page 30, lines 1 to 2.

Reviewer 2 Report
Comments and Suggestions for Authors
The review entitled “Revolutionizing Eye Care: Exploring the Potential of Microneedle Drug Delivery” is written very lengthy and needs to be trimmed to a very informative version. The authors covered the out-of-scope areas of the title presented. For ex., in abstract, divided into three sections and third sections written as general statement like application of microneedles in different modalities delivery.
Can the author please explain how the intravitreal injections delivery mode has low bioavailability in ocular delivery?
In line 28, the review focused on ocular delivery. What is the reason for discussing to be targeted to organs. The eye only has two segments. Please rewrite the statement.
Overall, the abstract needs to be revised in a brief, and concise as per journal guidelines.
Introduction
Line#5, it’s not ophthalmic conditions, it should be segments of barriers of the eye.
Please delete ‘Normal eye’ word from Figure 1.2.
What is the reason for focusing on only three diseases of the eye. Keratitis disease has lot of clinical benefits through microneedle technology application. Write a note on it.
Section 2 MN design and fabrication related to ocular application – is reported by number of publications. Can authors cite those references and delete this section.
https://www.ncbi.nlm.nih.gov/pmc/articles/PMC9623140/
https://www.sciencedirect.com/science/article/pii/S0169409X23003976.
Comments on the Quality of English Language
Moderate editing of English language required
Author Response
Response to Reviewer Comments
We are incredibly grateful to the expert reviewers for dedicating their time to reviewing our manuscript and providing encouraging feedback thoroughly. Their reviews and suggestions have greatly assisted in enhancing the clarity and quality of this manuscript. We have carefully addressed each comment raised by the reviewers individually. To easily keep track of the changes and corrections, we have highlighted them in yellow in the revised manuscript.
Reviewer #2
Comment 1: In abstract, divided into three sections and third sections written as general statement like application of microneedles in different modalities delivery. Can the author please explain how the intravitreal injections delivery mode has low bioavailability in ocular delivery? In line 28, the review focused on ocular delivery. What is the reason for discussing to be targeted to organs. The eye only has two segments. Please rewrite the statement. Overall, the abstract needs to be revised in a brief, and concise as per journal guidelines.
Response: Thank you for your valuable suggestion. The abstract has been revised as per your suggestions.
Comment 2: In the introduction, Line#5, it’s not ophthalmic conditions; it should be segments of barriers of the eye.
Response: Thank you for your valuable suggestion. Based on the reviewer's feedback, we have revised the sentence “Segments of barriers of the eye are classified into two distinct segments……”
Comment 3: Please delete the ‘Normal eye’ word from Figure 1.2.
Response: Thank you for your valuable suggestions; as per the suggestion, we deleted the word normal eye from Figure 1.2. in the revised manuscript (page 3).
Comment 4: What is the reason for focusing on only three diseases of the eye. Keratitis disease has lot of clinical benefits through microneedle technology applications. Write a note on it.
Response: Thank you for your valuable suggestion. As per the reviewer's suggestion, we have included other diseases in section 5.4, on page 42, lines 17-52, page 43, lines 1-52, and page 44, lines 1-7.
Comment 5: Section 2 MN design and fabrication related to ocular application – is reported by number of publications. Can authors cite those references and delete this section?
https://www.ncbi.nlm.nih.gov/pmc/articles/PMC9623140/
https://www.sciencedirect.com/science/article/pii/S0169409X23003976.
Response: Thank you for your suggestion. We appreciate your input and the references provided. This is a comprehensive review, so the section on MN design and fabrication related to ocular applications is essential. Including this information will provide readers with a thorough understanding of the topic within the same article, enhancing the quality and completeness of the manuscript. Therefore, we prefer to retain this section while citing the relevant references to support our discussion, page 10, references 47-48.

Reviewer 3 Report
Comments and Suggestions for Authors
I have the following suggestion for the manuscript
1. Abstract is too long. It should be concise and to the point of investigation.
2. Introduction needs to be focused on the area which is covered in this review.
3. It is very becessary to provide at least three graphical explanation of the different mechanism
4. The future prospects needs to be incorporated and evaluation of this technology in term of Pharm industry.
5. It would be appreciated if marketed patches table is included
Comments on the Quality of English LanguageEnglish is okay
Author Response
Response to Reviewer Comments
We are incredibly grateful to the expert reviewers for dedicating their time to reviewing our manuscript and providing encouraging feedback thoroughly. Their reviews and suggestions have greatly assisted in enhancing the clarity and quality of this manuscript. We have carefully addressed each comment raised by the reviewers individually. To easily keep track of the changes and corrections, we have highlighted them in yellow in the revised manuscript.
Reviewer #3
Comment 1: Abstract is too long. It should be concise and to the point of investigation.
Response: Thank you for your valuable suggestions. We have revised the abstract according to the reviewer's suggestion
.Comment 2: The introduction needs to be focused on the area which is covered in this review.
Response: Thank you for your valuable suggestions. Per the reviewers' feedback, we revised the introduction to focus on the manuscript's interest and objective.
Comment 3: It is very necessary to provide at least three graphical explanations of the different mechanism
Response: Thank you for your valuable suggestion. We have included a graphical explanation for the different mechanisms of drug release from microneedles in Figure 3b, page 11.
Comment 4: The future prospects need to be incorporated and evaluation of this technology in terms of the Pharm industry.
Response: Thank you for your valuable suggestions; we have already included a separate section on “. Section 7. Challenges and future directions” and industrial requirements for this product development and evaluation in this section.
Comment 5: It would be appreciated if marketed patches table is included.
Response: Thank you for your valuable suggestion. As per the reviewer's suggestion, we have included the microneedle-enabled marketed patches, Table 9, page 51.

Reviewer 4 Report
Comments and Suggestions for Authors
The paper is an extensive review on microneedles applied to eye healthcare. Even if of interest, the paper should be strongly revised before being considered for publication. The Abstract is too long and it should be shortened. In general, the paper is too long and it should be made more easy readable, otherwise it is almost impossible to find useful information among so much words. All figures are of very bad quality and they should be improved. More important are the captions that must well describe in detail what is reported in the figures. All the captions are too much synthetic, failing in describing the figures. Table 1 and Table 2 should report materials and characteristics only of microneedles used in eye helathcare, and not for every kind of microneedles, since this information is redundant with respect to the argument of the paper. In the subchapter 2.2.1, the papers from Dardano, Principia, et al. "A photolithographic approach to polymeric microneedles array fabrication." Materials 8.12 (2015): 8661-8673, and Dardano, Principia, et al. "One-shot fabrication of polymeric hollow microneedles by standard photolithography." Polymers 13.4 (2021): 520. should be cited for the sake of completeness. For Table 3 and 4, the same holds as Table 1 and 2. There is not any need to divide subchapter 2.5 in so many sub-subchapters! Figure 6 can be cancelled, since it is only an elementary sketch. It is not clear the significance of Figure 7 and 8 that could be cancelled since they are not explicative of the use of microneedles in eye pathologies. Many errors are wrongly written. errors must be always be written with only one significant digit (at least two if the first is 1), even when of statistical nature. Please, check the errors formati in the text and in the tables.
Comments on the Quality of English LanguageEnglish style and format should be revised.
Author Response
Response to Reviewer Comments
We are incredibly grateful to the expert reviewers for dedicating their time to reviewing our manuscript and providing encouraging feedback thoroughly. Their reviews and suggestions have greatly assisted in enhancing the clarity and quality of this manuscript. We have carefully addressed each comment raised by the reviewers individually. To easily keep track of the changes and corrections, we have highlighted them in yellow in the revised manuscript.
Reviewer 4
Comment 1: The Abstract is too long and it should be shortened. In general, the paper is too long and it should be made more easy readable, otherwise it is almost impossible to find useful information among so much words.
Response: The author sincerely thanks you for your valuable comments and suggestions. As per the reviewer's suggestion, we thoroughly revised the abstract and manuscript.
Comment 2: The captions of figures must well describe in detail what is reported in the figures. All the captions are too much synthetic, failing in describing the figures.
Response: Thank you for your valuable suggestions. We have revised all the figure captions and provided detailed captions in the revised manuscript.
Comment 3: Table 1 and Table 2 should report materials and characteristics only of microneedles used in eye helathcare, and not for every kind of microneedles, since this information is redundant with respect to the argument of the paper.
Response: Thank you for your thoughtful feedback. We have carefully considered the reviewer's comments and made revisions to Tables 1 and 2. We have included only the information specifically related to ophthalmic microneedles and have cited the relevant references. The revised tables 1 and 2 can be located on pages 13-14 and 15-16.
Comment 4: In subchapter 2.2.1, the papers from Dardano, Principia, et al. "A photolithographic approach to polymeric microneedles array fabrication." Materials 8.12 (2015): 8661-8673, and Dardano, Principia, et al. "One-shot fabrication of polymeric hollow microneedles by standard photolithography." Polymers 13.4 (2021): 520. should be cited for the sake of completeness.
Response: Thank you for the valuable suggestion. We have referenced two important papers to improve the quality of the manuscript. We cited these two papers in section 2.1 Table 1, references 84 and 85.
Comment 5: For Table 3 and 4, the same holds as Table 1 and 2.
Response: Thank you for your insightful observations. We agree with the reviewer's comments and have revised tables 3 and 4. We have included only the information specific to ophthalmic microneedles and have referenced related sources. The revised tables 3 and 4 can be found on pages 19 and 21.
Comment 6: There is not any need to divide subchapter 2.5 in so many sub-subchapters!
Response: Thank you for your comments; we removed the subsection from section 2.5 in the revised manuscript.
Comment 7: Figure 6 can be cancelled, since it is only an elementary sketch. It is not clear the significance of Figure 7 and 8 that could be cancelled since they are not explicative of the use of microneedles in eye pathologies.
Response: Thank you for your valuable comment. We have included this figure to enhance the quality and understanding of the manuscript. We believe these figures are crucial for new researchers to comprehend the manuscript.
Comment 8: Many errors are wrongly written. errors must be always be written with only one significant digit (at least two if the first is 1), even when of statistical nature. Please, check the errors formati in the text and in the tables.
Response: Thank you for your valuable comment. We have thoroughly reviewed and revised the errors in the manuscript.

Round 2
Reviewer 2 Report
Comments and Suggestions for Authors
Thank you for addressing the comments.